**Perspectives**

# A general and efficient representation of ancestral recombination graphs

Yan Wong (ID) ,[1] Anastasia Ignatieva (ID) ,[2,3] Jere Koskela (ID) ,[4,5] Gregor Gorjanc (ID) ,[6] Anthony W. Wohns (ID) ,[7,8] Jerome Kelleher (ID) [1,*]

[1]Big Data Institute, Li Ka Shing Centre for Health Information and Discovery, University of Oxford, Oxford OX3 7LF, UK
[2]School of Mathematics and Statistics, University of Glasgow, Glasgow G12 8TA, UK
[3]Department of Statistics, University of Oxford, Oxford OX1 3LB, UK
[4]School of Mathematics, Statistics and Physics, Newcastle University, Newcastle NE1 7RU, UK
[5]Department of Statistics, University of Warwick, Coventry CV4 7AL, UK
[6]The Roslin Institute and Royal (Dick) School of Veterinary Studies, University of Edinburgh, Edinburgh EH25 9RG, UK
[7]Broad Institute of MIT and Harvard, Cambridge, MA 02142, USA
[8]Department of Genetics, Stanford University School of Medicine, Stanford, CA 94305-5101, USA

*Corresponding author: Big Data Institute, Li Ka Shing Centre for Health Information and Discovery, University of Oxford, Oxford OX3 7LF, UK. Email: jerome.kelleher@bdi.ox.ac.uk

As a result of recombination, adjacent nucleotides can have different paths of genetic inheritance and therefore the genealogical trees for a sample of DNA sequences vary along the genome. The structure capturing the details of these intricately interwoven paths of inheritance is referred to as an ancestral recombination graph (ARG). Classical formalisms have focused on mapping coalescence and recombination events to the nodes in an ARG. However, this approach is out of step with some modern developments, which do not represent genetic inheritance in terms of these events or explicitly infer them. We present a simple formalism that defines an ARG in terms of specific genomes and their intervals of genetic inheritance, and show how it generalizes these classical treatments and encompasses the outputs of recent methods. We discuss nuances arising from this more general structure, and argue that it forms an appropriate basis for a software standard in this rapidly growing field.

Keywords: ancestral recombination graphs

## Introduction

Estimating the genetic genealogy of a set of DNA sequences under the influence of recombination, usually known as an ancestral recombination graph (ARG), is a long-standing goal in genetics. Broadly speaking, an ARG describes the different paths of genetic inheritance caused by recombination, encapsulating the resulting complex web of genetic ancestors of a set of sampled genomes. Recent breakthroughs in large-scale inference methods (Rasmussen *et al.* 2014; Kelleher *et al.* 2019b; Speidel *et al.* 2019; Schaefer *et al.* 2021; Wohns *et al.* 2022; Zhan *et al.* 2023; Zhang *et al.* 2023; Deng *et al.* 2024) have raised the realistic prospect of ARG-based analysis becoming a standard part of the population and statistical genetics toolkit (Hejase *et al.* 2020). Applications using inferred ARGs as input have begun to appear (Osmond and Coop 2021; Fan *et al.* 2022; Guo *et al.* 2022; Hejase *et al.* 2022; Fan *et al.* 2023; Ignatieva *et al.* 2023; Link *et al.* 2023; Nowbandegani *et al.* 2023; Zhang *et al.* 2023; Deraje *et al.* 2024; Grundler *et al.* 2024; Huang *et al.* 2024; Korfmann *et al.* 2024) and many more are sure to follow (Harris 2019, 2023).

Although it is widely accepted that ARGs are important, there is some confusion about what, precisely, an ARG *is*. In its original form, developed by Griffiths and colleagues, the ARG is an alternative formulation of the coalescent with recombination (Hudson 1983a), where the stochastic process of coalescence

and recombination among ancestral lineages is formalized as a graph (Ethier and Griffiths 1990; Griffiths 1991; Griffiths and Marjoram 1996, 1997). Subsequently, an ARG has come to be thought of as a data structure (Minichiello and Durbin 2006), i.e. describing a *realization* of such a random process, or an inferred ancestry of a sample of genomes. The distinction between stochastic process and data structure is not clear cut, however, and subfields use the term differently (Appendices A and B). The term "ARG," therefore, has many different meanings, varying over time and depending on context. There is, however, an emerging consensus to use the term in quite a general sense (e.g. Hejase *et al.* 2020; Mathieson and Scally 2020; Schaefer *et al.* 2021; Fan *et al.* 2023; Harris 2023; Zhang *et al.* 2023), informally encompassing the varied structures output by modern simulation and inference methods (Rasmussen *et al.* 2014; Palamara 2016; Haller *et al.* 2019; Kelleher *et al.* 2019b; Speidel *et al.* 2019; Baumdicker *et al.* 2022; Zhang *et al.* 2023). There is currently no formal definition or systematic discussion that unifies these different structures, however, stifling progress in this vibrant research area.

In this perspective, we provide a simple formal definition of an ARG data structure which generalizes classical definitions and encompasses the output of modern simulation and inference methods. We show that different levels of approximation are possible using this structure, illustrated via examples. The proposed ARG definition is the basis of the widely used `tskit` library which

provides a powerful software platform for ARG-based analysis and, we argue, would be a useful community standard. This perspective is intended for "ARG practitioners," who we hope will find the detailed examples, technical appendices, and comprehensive bibliography useful. Readers seeking an introduction to ARGs and their applications are directed to Lewanski *et al.* (2024) and Brandt *et al.* (2024).

## Genome ARGs

We define a genome as the complete set of genetic material that a child inherits from 1 parent. A diploid individual, therefore, carries 2 genomes, 1 inherited from each parent (we assume diploids and consider nuclear autosomal DNA here for clarity, but the definitions apply to organisms of arbitrary ploidy). We will also use the term "genome" in its more common sense of "the genome" of a species, and hope that the distinction will be clear from the context. We are not concerned here with mutational processes or observed sequences, but consider only processes of inheritance, following the standard practice in coalescent theory. We also do not consider structural variation, and assume that all samples and ancestors share the same genome coordinate space.

A genome ARG (gARG) is a directed acyclic graph in which nodes represent haploid genomes and edges represent genetic inheritance between an ancestor and a descendant. The topology of a gARG specifies that genetic inheritance occurred between ancestors and descendants, but the graph connectivity does not tell us which *parts* of their genomes were inherited. In order to capture the effects of recombination we "annotate" the edges with the genome coordinates over which inheritance occurred. This is sufficient to describe the effects of inheritance under any form of homologous recombination (such as multiple crossovers during a single round of meiosis, gene conversion events, and many forms of bacterial and viral recombination).

We can define a gARG formally as follows. Let $N = \{1, \ldots, n\}$ be the set of nodes representing the genomes in the gARG, and $S \subseteq N$ be the set of sampled genomes. Then, $E$ is the set of edges, where each element is a tuple $(c, p, I)$ such that $c, p \in N$ are the child and parent nodes and $I$ is the set of disjoint genomic intervals over which genome $c$ inherits from $p$. Thus, each topological connection between a parent and child node in the graph is annotated with a set of inheritance intervals $I$. Here, the terms parent and child are used in the graph sense; these nodes, respectively, represent ancestor and descendant genomes, which can be separated by multiple generations. We will use these 2 sets of terms interchangeably.

How nodes are interpreted, exactly, is application dependent. Following Hudson (1983a), we can view nodes as representing gametes, or we can imagine them representing, for example, the genomes present in cells immediately before or after some instantaneous event (Appendix D). A node can represent any genome along a chain of cell divisions or can be interpreted as representing one of the genomes of a potentially long-lived individual. In many settings, nodes are dated, i.e. each node $u \in N$ is associated with a time $\tau_u$, and how we assign precise times will vary by application. The topological ordering defined by the directed graph structure and an arrow of time (telling us which direction is pastwards) is sufficient for many applications, however, and we assume node dates are not known here. In practical settings, we will wish to associate additional metadata with nodes such as sample identifiers or quality-control metrics. It is, therefore, best to think of the integers used here in the definition of a node as an *identifier*, with which arbitrary additional information can be associated.

As illustrated in Fig. 1, the gARG for a given set of individuals is embedded in their pedigree. The figure shows the pedigree of 8 diploid individuals and their 16 constituent genomes (each consisting of a single chromosome), along with paths of genetic inheritance. Here, and throughout, nodes are labeled with uppercase alphabetical letters rather than integer identifiers to avoid confusion with genomic intervals. Thus individual $d_1$ is composed of genomes $A$ and $B$, which are inherited from its 2 parents $d_3$ and $d_4$. Each inherited genome may be the recombined product of the 2 genomes belonging to an individual parent. In this example, genome $B$ was inherited directly from $d_4$'s genome $G$ without recombination, whereas genome $A$ is the recombinant product of $d_2$'s genomes $E$ and $F$ crossing over at position 2. Specifically, genome $A$ inherited the (half-closed) interval $[0, 2)$ from genome $E$ and $[2, 10)$ from genome $F$. These intervals are shown attached to the corresponding graph edges. The figure shows the annotated pedigree with realized inheritance of genomes between generations (a), the corresponding gARG (b), and finally the corresponding sequence of local trees along the genome (c). The local trees span the 3 genome regions delineated by the 2 recombination breakpoints that gave rise to these genomes; see Appendix E for details on how local trees are embedded in an ARG.

## Event ARGs

A classical view of an ARG data structure, described explicitly in several publications (e.g. Wiuf and Hein 1999b; Gusfield 2014; Hayman *et al.* 2023), interprets nodes not as genomes but as historical *events* (but see Parida *et al.* 2011 and Zhang *et al.* 2023 for notable exceptions). This event ARG (eARG) encoding is the basis of the output formats created by multiple ARG inference tools (e.g. Song and Hein 2004; Song *et al.* 2005; Rasmussen *et al.* 2014; Heine *et al.* 2018; Ignatieva *et al.* 2021). In this encoding, there are 2 types of internal node in the graph, representing the most recent common ancestor and recombination events in the history of a sample. At common ancestor nodes, the inbound lineages merge into a single ancestral lineage with 1 parent, and at recombination nodes a single lineage is split into 2 independent ancestral lineages. Recombination nodes are annotated with the corresponding crossover breakpoints, and these breakpoints are used to construct the local trees. This is done by tracing pastwards through the graph from the samples, making decisions about which outbound edge to follow through recombination nodes based on the breakpoint position (Griffiths and Marjoram 1996). Figure 2 shows an example of an eARG with 3 sample genomes ($A$, $B$, and $C$), 3 common ancestor events ($E$, $F$, and $G$) and a single recombination event (node $D$) with a breakpoint at position $x$. Assigning a breakpoint to a recombination node is not sufficient to uniquely define the local trees, and either some additional ordering rules (e.g. Griffiths and Marjoram 1996) or explicit information (e.g. Gusfield 2014; Ignatieva *et al.* 2021) is required to distinguish the left and right parents. We assume in Fig. 2 that $D$ inherits genetic material to the left of $x$ from $E$ and to the right of $x$ from $F$.

While this approach of annotating recombination nodes with a breakpoint in an eARG is a concise and elegant way of describing realizations of the coalescent, it has limitations. The eARG encoding explicitly models only 2 different types of event; thus anything that is not a single crossover recombination or common ancestor event must be incorporated either in a roundabout way using these events, or by adding new types of event to the encoding. For example, gene conversion (Wiuf and Hein 2000) could be accommodated either by stipulating a third type of event (annotated

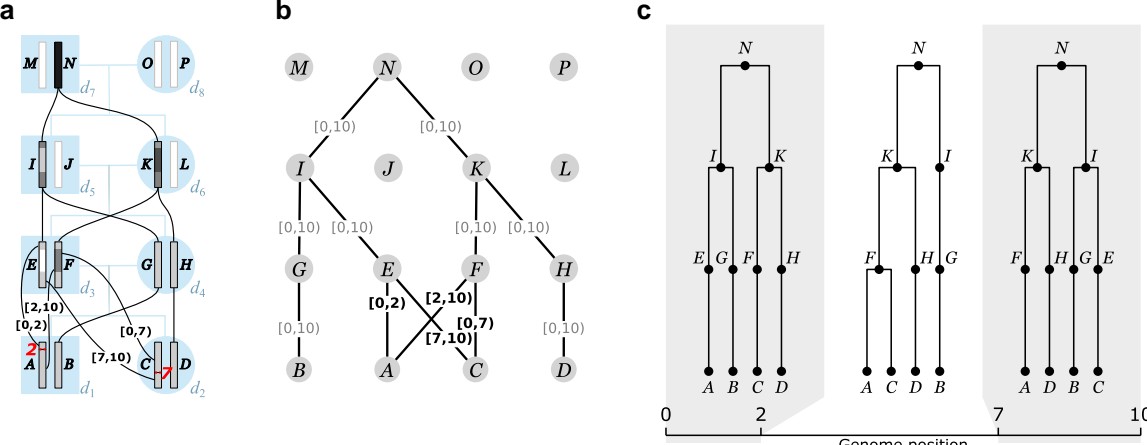

**Fig. 1.** An example gARG embedded in a pedigree. **a.** Diploid individuals (shaded backgound squares / circles), visualized in a highly inbred pedigree and labeled $d_1$–$d_8$, contain both paternal and maternal genomes labeled *A*–*P*. Black lines show inheritance paths connecting genomes in the current generation (*A*–*Dd*) with their ancestors. Genomes *A*–*C* are the product of 2 independent meioses (recombination events, with italicised breakpoint positions) between the paternal genomes *E* and *F*, and regions of genome inherited are shown with shaded color. Genomes are shaded such that where, backwards in time, they merge into a common ancestor, the merged region is darker. **b.** The corresponding gARG along with inheritance annotations on all edges (partial inheritance in bold). **c.** The corresponding local trees.

by 2 breakpoints and corresponding traversal conventions for recovering the local trees) or by 2 recombination nodes joined by a zero-length edge. The gARG encoding described in the previous section offers a simpler and more direct solution.

Aside from these practical challenges, there is also a deeper issue with the implicit strategy of basing an ARG data structure on recording events and their properties (e.g. the crossover breakpoint for a recombination event). This approach requires all events to be recorded explicitly, and does not provide an obvious mechanism for aggregating multiple, potentially unresolvable, events. As datasets approach the population scale (e.g. Bycroft *et al.* 2018; Turnbull *et al.* 2018; Hayes and Daetwyler 2019; Karczewski *et al.* 2020; Ros-Freixedes *et al.* 2020; Tanjo *et al.* 2021; Halldorsson *et al.* 2022) representing such uncertainty directly through the data structure is a useful alternative to classical methods based on probabilistic sampling.

## Ancestral material and sample resolution

Ancestral material (Wiuf and Hein 1999a, 1999b) is a key concept in understanding the overall inheritance structure of an ARG. It denotes the genomic intervals ancestral to a set of samples on the edges of an ARG. For example, in Fig. 1, we have 4 sample genomes, *A*–*D*. As we trace their genetic ancestry into the previous generation (*E*–*H*), we can think of their ancestral material propagating through the graph pastwards. In the region [2, 7), there is a local coalescence where nodes *A* and *C* find a common ancestor in *F*. Thus, in this region, we have 3 genome segments that are ancestral to the 4 samples. Figure 1a illustrates this by (shaded) ancestral material being present in only 3 nodes (*F*, *G*, and *H*) in this region, while node *E* is blank as it carries *nonancestral* material. This process of local coalescence continues through the graph, until all samples reach their most recent common ancestor in node *N*.

The process of tracking local coalescences and updating segments of ancestral material is a core element of Hudson's seminal simulation algorithm (Hudson 1983b; Kelleher *et al.* 2016). The ability to store resolved ancestral material is also a key distinction between the eARG and gARG encodings. Because an eARG stores only the graph topology and recombination breakpoints, there is

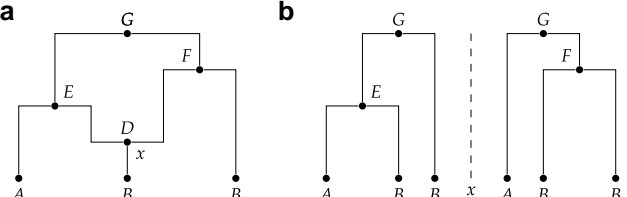

**Fig. 2.** A classical eARG. **a.** Standard graph depiction with breakpoint *x* associated with the recombination node *D*. Nodes *E*, *F*, and *G* are common ancestor events. **b.** Corresponding local trees to the left and right of breakpoint *x* (note these are shown in the conventional form in which only coalescences within the local tree are included hence *D* is omitted; see Appendix E for a discussion of this important point).

no way to locally ascertain ancestral material without traversing the graph pastwards from the sample nodes, resolving the effects of recombination and common ancestor events.

Efficiently propagating and resolving ancestral material for a sample through a preexisting graph is a well-studied problem, and central to recent advances in individual-based forward-time simulations (Haller et al. 2019; Kelleher et al. 2018). In contrast to the usual "retrospective" view of ARGs discussed so far, these methods record an ARG forwards in time in a "prospective" manner. Genetic inheritance relationships and mutations are recorded exhaustively, generation-by-generation, leading to a rapid buildup of information, much of which will not be relevant to the genetic ancestry of a future population. This redundancy is periodically removed using the "simplify" algorithm (Kelleher et al. 2018), which propagates and resolves ancestral material. Efficient simplification is the key enabling factor for this prospective-ARG-based approach to forward-time simulation, which can be orders of magnitude faster than standard sequence-based methods (see Appendix G for other applications of ARG simplification). We refer to a gARG that has been simplified with respect to a set of samples, such that the inheritance annotations on its edges contain no nonancestral material, as sample-resolved.

Any eARG can be converted to a sample-resolved gARG via a 2-step process illustrated in Fig. 3. The first step is to take the input eARG (Fig. 3a), duplicate its graph topology, and then add inheritance annotations to each of the gARG's edges (Fig. 3b) as follows. If

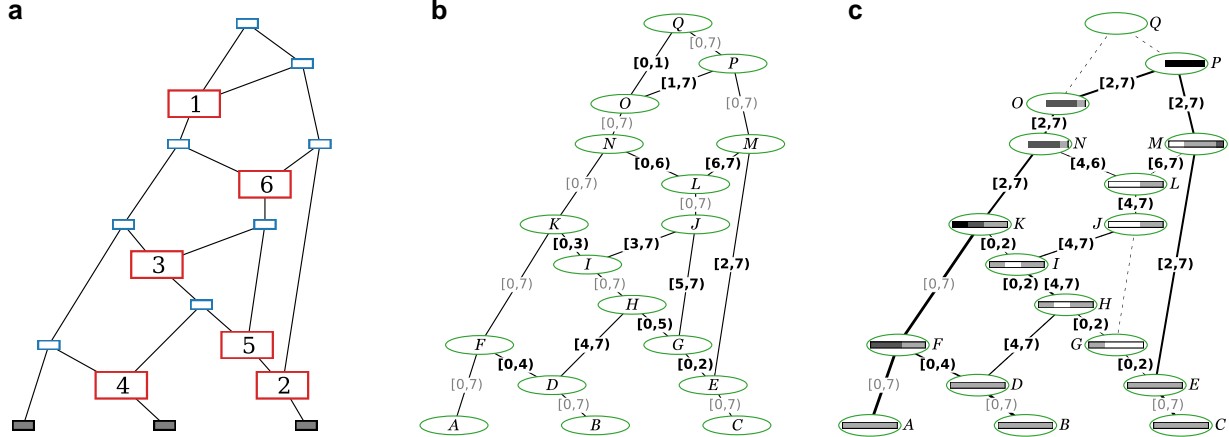

**Fig. 3.** Converting the Wiuf and Hein (1999b, Fig. 1) example to a sample-resolved gARG. **a.** The original eARG; nodes represent sampling, common ancestor, and recombination events (small shaded, small blue, and large red rectangles respectively); the latter contain breakpoint positions. **b.** The corresponding gARG with breakpoints directly converted to edges annotated with inheritance intervals. **c.** The sample-resolved gARG resulting from simplifying with respect to the sample genomes, A, B, and C. Dashed lines show edges that are no longer present (in practice, nodes G, J, and Q would also be removed). Coalescence with respect to the sample is indicated by shaded bars, as in Fig. 1a; nodes N, O, P, Q have truncated bars showing that local ancestry of entirely coalesced regions is omitted. Line thickness is proportional to the genomic span of each edge. Nodes representing recombination events are retained for clarity, but could be removed by simplification if desired.

a given node is a common ancestor event, we annotate the single outbound edge with the interval $[0, L)$ , for a genome of length $L$. If the node is a recombination event with a breakpoint $x$, we annotate the 2 outbound edges, respectively, with the intervals $[0, x)$ and $[x, L)$. These inheritance interval annotations are clearly in one-to-one correspondence with the information in the input eARG. They are also analogous to the inheritance intervals we get on the edges in a prospective gARG produced by a forward-time simulation, which are concerned with recording the direct genetic relationship between a parent and child genome and are not necessarily minimal in terms of the ancestral material of a sample. Thus, the final step is to use the "simplify" algorithm to re-solve the ancestral material of the samples (Fig. 3c).

The sample-resolved gARG of Fig. 3c differs in some important ways to the original eARG (Fig. 3a). Firstly, we can see that some nodes and edges have been removed entirely from the graph. The "grand MRCA" Q is omitted from the sample-resolved gARG because all segments of the genome have fully coalesced in K and P before Q is reached. Likewise, the edge between G and J is omitted because the recombination event at position 5 (represented by node G) fell in nonancestral material. More generally, we can see that the sample-resolved gARG of Fig. 3c allows for "local" inspection of an ARG in a way that is not possible in an eARG. Because the ancestral material is stored with each edge of a gARG, the cumulative effects of events over time can be reasoned about, without first "replaying" those events. Many computations that we wish to perform on an ARG will require resolving the ancestral material with respect to a set of samples. The gARG encoding allows us to perform this once and to store the result, whereas the eARG encoding requires us to repeat the process each time.

## A diversity of structures

A key goal of this perspective is to highlight the heterogeneity of the graph structures inferred by modern ARG inference methods. To illustrate this point, Fig. 4 shows the output of KwARG (Ignatieva *et al.* 2021), ARGweaver (Rasmussen *et al.* 2014), tsinfer (Kelleher *et al.* 2019b), and Relate (Speidel *et al.* 2019) on the classical (Kreitman 1983) dataset. The ARGs in Fig. 4a and b are precise estimates (Appendix H), with each node corresponding to a common

ancestor or recombination event, or equivalently, either having 2 children or 2 parents. In contrast the ARGs in Fig. 4c and d do not have this clear-cut interpretation, and the nodes can simultaneously have more that than 2 children and more than 2 parents. Another dimension of variability among the ARGs is that the first 3 methods infer nodes that have a "coalescence span" greater than 0 and less than 100%, indicating that there are nodes that are "locally unary" (Appendix F), but mark a coalescence between lineages elsewhere along the sequence.

A key feature of the gARG encoding is that it enables these varying levels of precision to be represented. These ideas are illustrated in Fig. 5, which shows different levels of "simplification" (Appendix G) of the same underlying simulated ARG. The full ARG, with all co-alescent and recombination events represented by separate genomes, is shown in Fig. 5a. Simpler representations can be formed by removing "unknowable" nodes such as those in singly connected graph components (Fig. 5b) and collapsing multiple recombinations into a single child or multiple coalescences into a single parent (Fig. 5c). Finally, Fig. 5d is a "fully simplified" ARG, in which only coalescences in local trees are retained. Note that while ARGs of this type (produced by default by the msprime simulator, for example) lack a significant level of detail, they still retain the key feature of shared node identity across local trees.

This ability to represent an ARG to differing degrees of precision is a powerful feature. In particular, when inferring ARGs from genome sequencing data, the timing, positions, and even the number of recombination events is generally not possible to infer precisely. For example, under coalescent-based models, the proportion of recombination events that change the ARG topology grows very slowly with sample size (Hein *et al.* 2004), and of those events only a small proportion are actually detectable from the data, assuming human-like mutation and recombination rates (Myers 2002; Hayman *et al.* 2023). Even when a recombination event is detectable, its timing and breakpoint position can only be inferred approximately, depending on how much information can be elucidated from mutations in the surrounding genomic region. A gARG can encode a diversity of ARG structures, including those where events *are* recorded explicitly, and those where they are treated as fundamentally uncertain and thus not explicitly inferred (Appendix H).

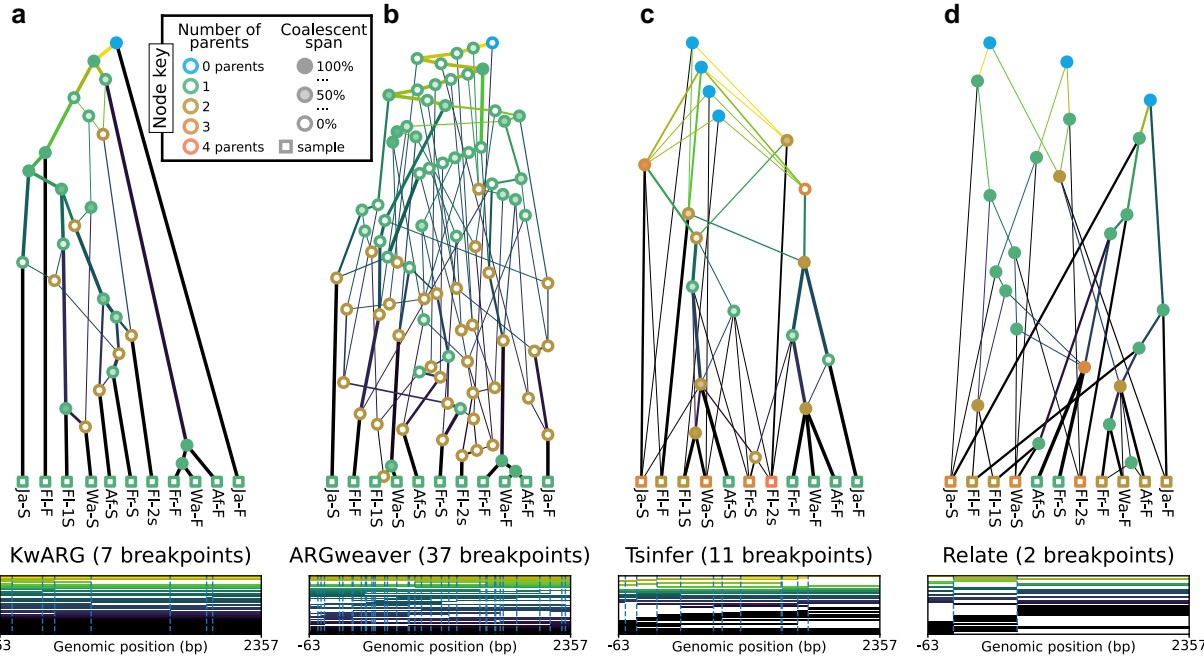

**Fig. 4.** Inference of sample-resolved ARGs for 11 *Drosophila melanogaster* DNA sequences over a 2.4 kb region of the ADH locus (Kreitman 1983). Results for 4 different methods: **a.** KwARG; **b.** ARGweaver; **c.** tsinfer; and **d.** Relate, converted to the standard *tskit* gARG encoding. See Appendix I for details of these methods. Edge colors indicate time of the edge's child node (lighter: older; darker: younger), with width proportional to genomic span. Vertical and horizontal positions of graph nodes are arbitrary. Bottom row graphics show the genome positions, relative to the start of the ADH gene, for each graph edge from the corresponding ARG. Edge intervals are drawn as horizontal lines, stacked in time order (edges with youngest children at the bottom); vertical dashed lines denote breakpoints between local trees.

## Implementation and efficiency

The gARG encoding can lead to highly efficient storage and processing of ARG data, and has been in use for several years. The succinct tree sequence data structure (usually known as a "tree sequence" for brevity) is a practical gARG implementation focused on efficiency. It was originally developed as part of the msprime simulator (Kelleher *et al.* 2016) and has subsequently been extended and applied to forward-time simulations (Haller et al. 2019; Kelleher *et al.* 2018), inference from data (Kelleher *et al.* 2019b; Wohns *et al.* 2022; Zhan *et al.* 2023), and calculation of population genetics statistics (Ralph *et al.* 2020). The succinct tree sequence encoding extends the basic definition of a gARG provided here by stipulating a simple tabular representation of nodes and edges, and also defining a concise representation of sequence variation using the "site" and "mutation" tables. The key property of the succinct tree sequence encoding that makes it an efficient substrate for defining analysis algorithms is that it allows us to sequentially recover the local trees along the genome very efficiently, and in a way that allows us to reason about the *differences* between those trees (Kelleher *et al.* 2016; Ralph *et al.* 2020).

The tskit library is a liberally licensed open-source toolkit that provides a comprehensive suite of tools for working with gARGs (encoded as a succinct tree sequence). Based on core functionality written in C, it provides interfaces in C, Python and Rust. Tskit is mature software, widely used in population genetics, and has been incorporated into numerous downstream applications (e.g. Haller and Messer 2019; Speidel *et al.* 2019; Adrion *et al.* 2020; Terasaki Hart *et al.* 2021; Baumdicker *et al.* 2022; Fan *et al.* 2022; Guo *et al.* 2022; Mahmoudi *et al.* 2022; Fan *et al.* 2023; Ignatieva *et al.* 2023; Korfmann *et al.* 2023; Nowbandegani *et al.* 2023; Petr *et al.* 2023; Rasmussen and Guo 2023; Tsambos *et al.* 2023; Zhang *et al.* 2023; Korfmann *et al.* 2024; Tagami *et al.* 2024). The technical details of

tskit, and how it provides an efficient and portable platform for ARG-based analysis, are beyond the scope of this manuscript.

## Discussion

Tremendous progress has been made in recent years on the long-standing problem of ARG inference, there is now a range of practically applicable methods available. Methods targeting large-scale datasets tend to simplify the inference problem by making a single, deterministic best-guess (Kelleher *et al.* 2019b; Speidel *et al.* 2019; Zhan *et al.* 2023; Zhang *et al.* 2023) (but see Deng *et al.* 2024 for recent developments in capturing uncertainty using a Bayesian framework, for relatively small sample sizes). Even under strict parsimony conditions and for small sample sizes the number of plausible ARGs compatible with a given dataset is vast, and it is, therefore, not clear that generating many guesses when sample sizes are large will achieve much in terms of capturing uncertainty. An alternative approach to is to incorporate uncertainty encountered during inference into the returned ARG. The gARG encoding described here enables particular kinds of uncertainty to be incorporated directly into the topology: nodes that have more than 2 children (polytomies) represent uncertainty over the ordering of coalescence events (Appendix D), and those that have more than 2 parents represent uncertainty over the ordering of multiple recombination events (Appendix G). Development of other methods to capture, for example, uncertainty about node ages and recombination breakpoint positions, is an important aspect of future work. How this uncertainty can be used in downstream applications is an open question.

Another important avenue for future work is to develop improved methods to evaluate and benchmark inference quality. In most cases, ARG inference is evaluated by simulating data

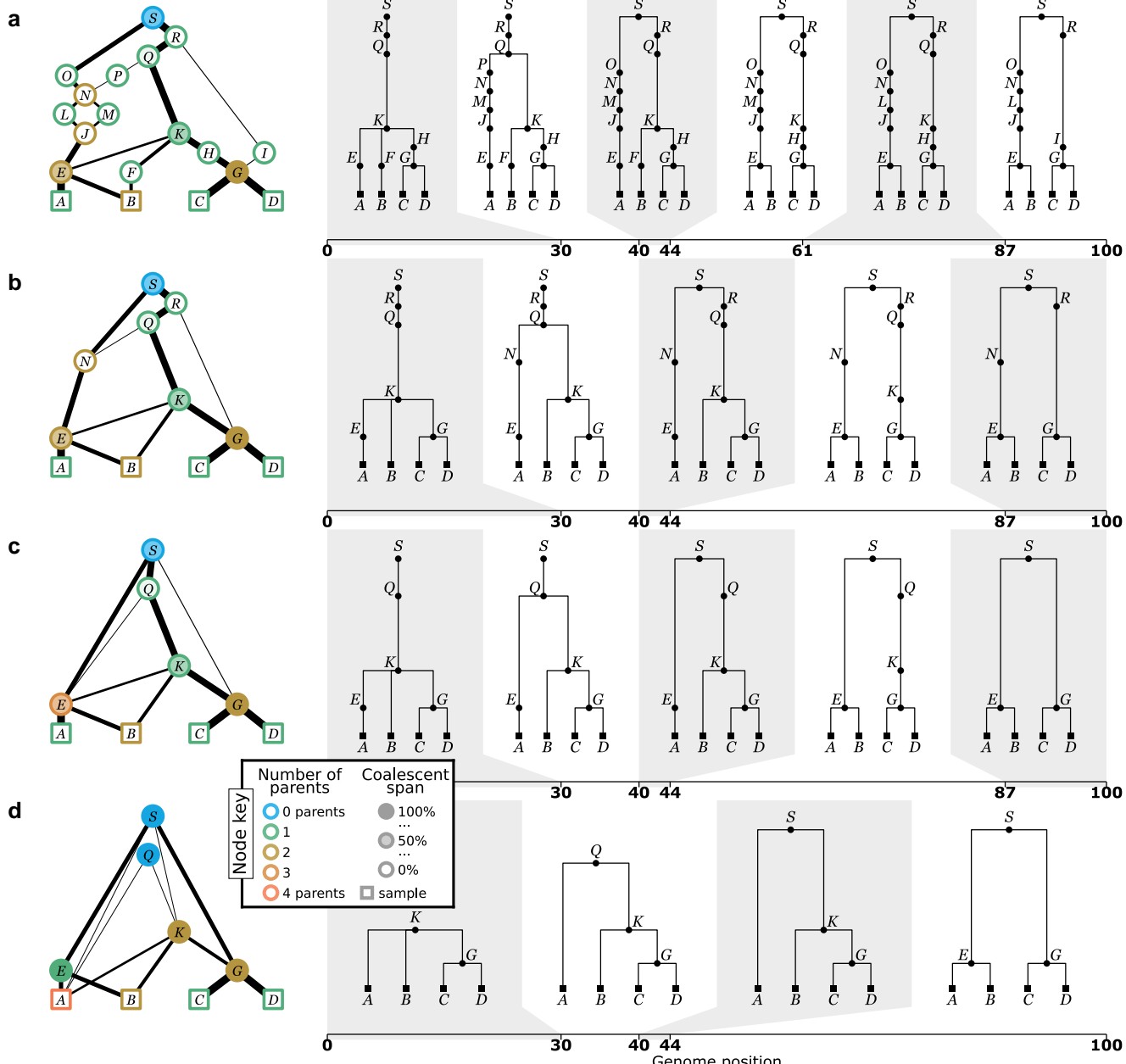

**Fig. 5.** Levels of ARG simplification. **a.** An example gARG simulated from a diploid Wright–Fisher model. **b.** Remove all singly connected graph components (e.g. diamonds such as *JLMN*). **c.** Remove nodes that never represent coalescences, i.e. are unary everywhere (e.g. *N* and *R*). **d.** Rewrite edges to bypass nodes in local trees in which they are unary (often described as "fully simplified"). In each case, the graph is shown on the left and corresponding local trees on the right. In the interest of visual clarity, inheritance intervals are not shown on the graph edges; Supplementary Fig. 1 shows the graphs with these inheritance intervals included. Graph nodes are colored by the number of parents and shaded according to the proportion of their span over which they are coalescent; see the text for more details.

from a known ground truth ARG, and comparing this to the inferred version via pairwise comparison of local trees along the genome using tree distance metrics (e.g. Robinson and Foulds 1981; Kendall and Colijn 2016), as described by Kuhner and Yamato (2015a). In comparing tree-by-tree along the genome, the effects of recombination are incorporated in an indirect manner through the correlations between the local trees, instead of directly taking into account the persistence of nodes and edges across multiple trees. The performance of tree distance metrics varies by application (Kuhner and Yamato 2015b), and the correct approach to handling subtleties such as polytomies is an open question (Kelleher *et al.* 2019b; Zhang *et al.* 2023). Tree distance

metrics often have $O(n^2)$ time complexity or worse and therefore cannot be applied to the very large sample sizes currently of interest. A recent trend has been to move away from such tree distance-based approaches and to examine more properties of the inferred ARGs, such as distributions of pairwise MRCA times (Brandt *et al.* 2022), waiting distances between local trees (Deng *et al.* 2021), and the genomic span of an edge or clade of samples (Ignatieva *et al.* 2023). In each case, simulation studies demonstrated substantial differences between these quantities in simulated and reconstructed ARGs that were not captured using tree-by-tree comparisons. Evaluations to-date have almost all been based on ground truth data from highly idealized

simulations, with sample sizes limited to at most a few thousand (typically much fewer). Beyond the effects of very simplistic error models (e.g. Kelleher *et al.* 2019b), the effects of the richness of real data at biobank-scale on ARG inference are almost entirely unknown. The development of ARG evaluation metrics that take into account more of the global topology and can be applied to large ARGs would be a valuable and timely addition to the field. Using ARGs simulated from observed pedigree data (Anderson-Trocmé *et al.* 2023) as ground-truth would also add a valuable dimension to our understanding of how well methods perform when faced with realistic population and family structure.

Interest in ARG inference methods and downstream applications is burgeoning, with exciting developments arriving at ever-increasing pace. Without agreement on basic terminology and some standardization on data formats, however, the ARG revolution may falter. For ARG-based methods to achieve mainstream status, we require a rich supporting software ecosystem. Ideally, this would comprise a wide range of inference methods specialized to different organisms, inference goals, and types and scales of data. If these diverse inference methods share a common, well-defined data format, their outputs could then be processed by many different downstream applications without the productivity-sapping problems of converting between partially incompatible formats (Excoffier and Heckel 2006). Earlier efforts to standardize ARG interchange shared this vision, but did not succeed (Cardona *et al.* 2008; McGill *et al.* 2013). Current methods tend to tightly couple both ARG inference and downstream analysis within the same software package, which is ultimately not compatible with the widespread use of ARGs for routine data analysis, and a healthy and diverse software ecosystem. The gARG encoding described here is a significant generalization of classical concepts, capable of describing even the bewildering complexity of contemporary datasets and encompassing a wide range of approximate ARG structures, and would be a reasonable basis for such a community interchange format.

Rigorously defining interchange formats (e.g. Kelleher *et al.* 2019a) is difficult and time-consuming, and no matter how precise the specification, in practise it is the *implementations* that determine how well methods interoperate. The BAM read alignment format (Li *et al.* 2009) is an instructive example. Originally developed as part of the 1000 Genomes project (1000 Genomes Project Consortium 2015) to address the fragmented software ecosystem that existed at the time (Danecek *et al.* 2021), BAM has since become ubiquitous in bioinformatics pipelines. The excellent interoperability between methods exchanging alignment data is largely attributable to the success of `htslib` (Bonfield *et al.* 2021), the software library that *implements* BAM and several other foundational bioinformatics file formats. Today, there are thousands of software projects using `htslib` (Bonfield *et al.* 2021), and it is this shared use of community software infrastructure that guarantees the smooth flow of data between applications. The emerging ARG software ecosystem could similarly benefit from the adoption of such shared community infrastructure to handle the mundane and time-consuming details of data interchange. The `tskit` library is a high-quality open-source gARG implementation, with proven efficiency and scalability (e.g. Anderson-Trocmé *et al.* 2023; Zhan *et al.* 2023), that is already in widespread use. Adopting it as a community standard may ease software implementation burden on researchers, freeing their time to address the many fascinating open questions and challenges that exist.

## Data availability

All code used to generate figures and run analyses is available on GitHub at https://github.com/tskit-dev/what-is-an-arg-paper. Supplemental material available at GENETICS online.

## Acknowledgments

We are grateful to Nick Barton, Gideon Bradburd, Castedo Ellerman, Alex Lewanski, Peter Ralph, and Andrew Vaughn for helpful discussions and comments on the manuscript, and to Iain Mathieson and Aylwyn Scally for interesting discussions that originally motivated this work. We are particularly grateful to Graham Coop for detailed feedback on an earlier version of the manuscript.

## Funding

G.G. acknowledges support from the BBSRC ISP grant to The Roslin Institute (BBS/E/D/30002275, BBS/E/RL/230001A, and BBS/E/RL/230001C). J.K. acknowledges support from the Robertson Foundation, NIH (research grants HG011395 and HG012473) and EPSRC (research grant EP/X024881/1). J.K. acknowledges support from EPSRC research grant EP/V049208/1.

## Conflicts of interest

The author(s) declare no conflicts of interest.

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

*Editor: G. Coop*

## Appendix A: Ancestral graphs: a brief history

The coalescent (Kingman 1982a, 1982b; Hudson 1983b; Tajima 1983) models the ancestry of a sample of genomes under an idealized population model, and provides the theoretical underpinning for much of contemporary population genetics. It is a stochastic process, where each random realization is a genealogical tree describing the genetic ancestry of the sample. Numerous extensions to the model have been proposed (Hudson 1990; Hein et al. 2004; Wakeley 2008), incorporating many evolutionary processes. Hudson (1983a) first incorporated recombination into the coalescent process, providing several fundamental analytical results and describing the basic simulation algorithm, still in widespread use (Hudson 2002; Kelleher et al. 2016; Kelleher and Lohse 2020; Baumdicker et al. 2022). In the 1990s, Griffiths and colleagues revisited the coalescent with recombination from a different perspective, formulating it as a stochastic process where each realization is encoded as a graph (Ethier and Griffiths 1990; Griffiths 1991; Griffiths and Marjoram 1996, 1997). They referred to both the stochastic process and its random realizations as the ancestral recombination graph (ARG). Although mathematically equivalent, it is important to note that the Griffiths and Hudson formulations of the coalescent with recombination are not identical; in particular, a direct implementation of the ARG process as originally described requires exponential time to simulate (see Appendix B for details). However, ARGs provided a way to reason about and infer recombinant ancestry as a single object, in a way that is not possible within Hudson's framework, which emphasized instead the collection of local trees along the genome resulting from recombination.

Subsequent work on ARGs proceeded in broadly 3 main directions: (1) exploring the mathematical properties of the coalescent with recombination and related stochastic processes; (2) inferring evolutionary parameters under (approximations to) this model, either with or without explicitly reconstructing the genealogy of the sample; and (3) treating the ARG as a discrete graph, ignoring the generating stochastic process, and studying its properties from a computational and algorithmic perspective.

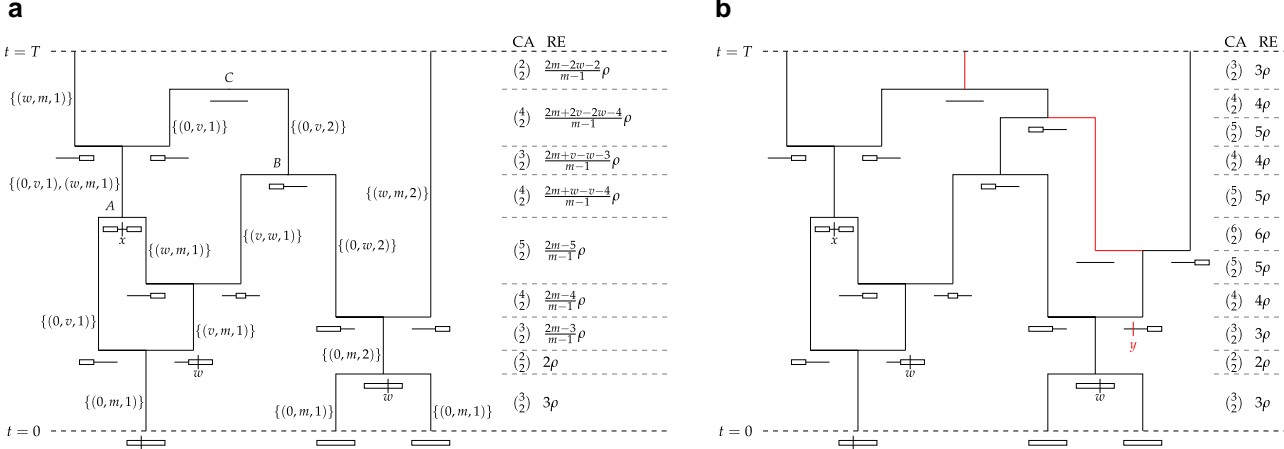

**Fig. A1. a.** A realization of the graph traversed by Hudson's algorithm started from a sample of 3 chromosomes with $m$ discrete sites each at time $t = 0$, and propagated until time $T$. The MRCA on the genetic interval $[v, w)$ is reached at event $B$, while that on $[0, v)$ is reached at event $C$. The nonancestral segment $[v, w)$ above a contributes to the rate of effective recombinations because it is trapped between ancestral segments. The 2 columns titled CA and RE are the respective rates of mergers and recombinations when the recombination rate is $\rho$. **b.** A corresponding realization of a Big ARG, which augments Hudson's algorithm by tracking nonancestral lineages. The result is a simpler state space and dynamics, at the cost of extra nodes and edges, highlighted in red, which do not affect the local tree at any site. Recombination positions are labeled alphabetically in time, and their ordering along the genome is $y < v < x < w$, of which the first only appears in panel b. There are 2 separate recombination events at link $w$.

An extensive body of work has been developed from studying the coalescent with recombination and other related graph-valued stochastic processes from a mathematical perspective. In particular, the Ancestral Selection Graph (ASG) (Krone and Neuhauser 1997; Neuhauser and Krone 1997) uses a similar approach to model natural selection instead of recombination. Unlike the ARG process, the ASG imposes a hard distinction between the stochastic process, which constructs a random ARG-like graph, and an observable realization, which is a single tree sampled from the graph in a nonuniform way to encode desired patterns of natural selection. Constructions of ASG-like stochastic processes encoding various forms of selection, often in parallel with recombination or other genetic forces, are an area of considerable and ongoing theoretical interest (e.g. Donnelly and Kurtz 1999; Neuhauser 1999; Fearnhead 2001, 2003; Etheridge and Griffiths 2009; Baumdicker and Pfaffelhuber 2014; González Casanova and Spanò 2018; Koskela and Wilke Berenguer 2019).

Early work on inference under the coalescent with recombination focused on the problem of inferring the parameters of the stochastic process, where the ancestry was regarded as a latent parameter to be averaged out (e.g. Griffiths and Marjoram 1996; Kuhner *et al.* 2000; Nielsen 2000; Fearnhead and Donnelly 2001). These methods met with limited success because the state space of ARGs is overwhelmingly large, and lacks a simple geometry or neighborhood structure for inference or sampling methods to exploit. Several breakthroughs in this direction were achieved through formulating simplified but more tractable approximations to the full model (McVean and Cardin 2005; Marjoram and Wall 2006; Li and Durbin 2011; Paul *et al.* 2011; Schiffels and Durbin 2014). The related problem of *sampling* genealogies compatible with a given dataset under the coalescent with recombination also proved notoriously difficult computationally; progress in explicitly inferring genealogies at scale has similarly been achieved through resorting to principled approximations (Rasmussen *et al.* 2014; Mahmoudi *et al.* 2022; Deng *et al.* 2024), or moving away from the coalescent with recombination altogether and seeking to infer a single plausible ARG (e.g.

Minichiello and Durbin 2006; Kelleher *et al.* 2019b; Speidel *et al.* 2019).

There has also been substantial interest in formulating and answering fundamental questions about properties of the ARG as a discrete graph structure, focusing on the ARG topology without considering either branch lengths or indeed the generating process. The first prominent problem was calculating (lower bounds on) the minimum number of recombinations required to reconstruct a valid genealogy for a given sample (Myers and Griffiths 2003), and constructing the corresponding minimal (parsimonious) ARGs (Song and Hein 2003; Lyngsø *et al.* 2005; Song *et al.* 2005). These problems are NP-hard in general (Wang *et al.* 2001), and progress has been achieved through studying various constrained special cases of ARGs (e.g. Gusfield *et al.* 2004) and other more general types of phylogenetic networks (Huson *et al.* 2010). The focus has been on algorithmic and combinatorial results (Gusfield 2014) that are often not of direct relevance to the inference problems described above.

The goal of this historical overview is to illustrate that the meaning of the term "ARG" now strongly depends on the context in which it is used, and can mean both the stochastic process that generates genealogies in the presence of recombination (e.g. Nordborg 2000; Birkner *et al.* 2013; Wilton *et al.* 2015; Griffiths *et al.* 2016), as well as the concrete realization of ancestry from a process (e.g. Gusfield 2014; Mathieson and Scally 2020; Brandt *et al.* 2022).

## Appendix B: The big and little ARG

Here, we review 2 important stochastic processes that construct ARGs: the "Big" ARG process of Griffiths and Marjoram (1997), and the "Little" ARG process of Hudson (1983a). The Big ARG process is mathematically simpler but is computationally intractable due to generating a vast number of ancestors which contribute no genetic material to the initial sample. The Little ARG process avoids nongenetic ancestors at the cost of more complex dynamics and state space. We also demonstrate that applications relying on the grouping of inheritance pathways into ancestral lineages,

such as likelihood-based inference under the coalescent, requires that the ARG data structure be interpreted in a model-specific way.

A generic state of the Little ARG process consists of a finite collection of lineages $L$, each of which is a list of disjoint ancestry segments $(\ell, r, a)$, where $[\ell, r)$ is a half-closed genomic interval and $a$ is an integer tracking the number of samples to which the lineage is ancestral over that interval. We also usually track the node associated with each segment, but that is not important for our purposes here so we omit it to lighten notation. The initial condition for a sample of $n$ genomes of length $m$ consists of $n$ lineages of the form $\{(0, m, 1)\}$. The process traverses a series of common ancestor and recombination events backwards in time. Recombination events happen at rate $\rho v/(m-1)$, where $\rho \geq 0$ is a per-genome recombination rate and

$$v = \sum_{x \in L}\left(\max_{(\ell,r,a) \in x} r - \min_{(\ell,r,a) \in x} \ell - 1\right)$$

is the number of available "links" surrounded by ancestral material. At a recombination event we choose one of these links uniformly and break it, replacing the original lineage in $L$ with 2 new lineages containing the ancestral material to the left and right of the break point, respectively.

Common ancestor events occur at rate $|L|2$. In a common ancestor event, 2 uniformly sampled lineages have their segments merged into a single ancestor lineage, which is added to $L$. If the lineages have overlapping intervals of ancestry, say, $(\ell, r, a_1)$ and $(\ell, r, a_2)$, a *coalescence* occurs. The result is a segment $(\ell, r, a_1 + a_2)$, and if $a_1 + a_2 < n$ it is included in the ancestor lineage. Otherwise, if $a_1 + a_2 = n$, we have found the most recent common ancestor of all samples in the interval $[\ell, r)$ and do not need to simulate its history any further. Nonoverlapping intervals from the 2 lineages are included in the ancestor lineage without changes. Eventually, we find resultant lineages in which all segments have fully coalesced, and so the number of extant lineages gradually falls to zero.

In the Big ARG process each edge in the graph corresponds to an extant lineage and nodes are events in the process. The $n$ initial leaf nodes are sampling events. Common ancestor events occur at rate $|L|2$. When a common ancestor event happens, 2 uniformly chosen lineages merge into a common ancestor lineage. Recombination events happen at rate $|L|\rho$. Here, we choose a lineage (i.e. edge) uniformly, and a breakpoint $0 < x < m$ uniformly on its genome. We terminate the edge at a node, record the breakpoint, and start 2 new edges from this node. The process then continues until there is only 1 lineage left (the grand most recent common ancestor, GMRCA), which is guaranteed to happen in finite time because of the quadratic rate of coalescing vs. linear rate of branching.

The state space of the Big ARG process is much simpler than that of the Little ARG process, which greatly facilitates mathematical reasoning. This simplicity comes at a substantial cost, however, if we wish to use it as a practical means of simulating recombinant ancestries. The number of events in the Big ARG all the way back to the GMRCA is $O(e^\rho)$ (Griffiths and Marjoram 1997), whereas the number of events required to simulate the Little ARG is $O(\rho^2)$ (Hein *et al.* 2004; Baumdicker *et al.* 2022). This disparity arises because the majority of the events in the Big ARG are recombination events which occur outside of ancestral material, and these do not have any bearing on the ancestry of the initial sample. Because we don't keep track of the distribution of ancestral material during the process, we generate a vastly larger graph.

Figure A1 illustrates the more complex state space of the Little ARG process, as well as the extra events which occur in the Big ARG process. Moreover, it depicts the rates of common ancestors and recombination events in each interval of time of the realizations. In order to evaluate these rates, e.g. for likelihood-based inference (Baumdicker *et al.* 2022; Mahmoudi *et al.* 2022), it is necessary to know the number of lineages and number of extant links available for recombination in each time interval. Some representations may not provide this information. For example, in the gARG encoding depicted in Fig. 3c, it is clear that a recombination takes place between nodes *I*, *K*, and *J*. But the exact time of the recombination event is ambiguous: it could take place at any time between node *I* and its parents and produce the same gARG. Because a recombination increases the number of extant lineages by 1 (in the rootward direction of time), the number of lineages during the same time interval is ambiguous as well. In fact, this information cannot be recovered from the gARG encoding used in Fig. 3c without an extrinsic convention. For the basic coalescent with recombination, it is sufficient to create 2 gARG nodes at the time of the recombination event, with the interpretation that the 2 rootward edges from node *I* in Fig. 3c belong to the same lineage until the time of nodes *K* and *J*, and split into 2 separate lineages at that time point. Similarly, the trapped, nonancestral links along that lineage remain available for effective recombination (i.e. one which splits up ancestral material) for the same time interval. This interpretation is highlighted in Fig. A1 by drawing only 1 vertical edge between a recombinant child and its 2 parents.

## Appendix C: Survey of ARG inference methods

The problem of reconstructing ARGs for samples of recombining sequences has been of interest since the ARG was first defined. Early methods focused on finding parsimonious ARGs, i.e. those with a minimal number of recombination events (Hein 1990). Two main approaches emerged: "backwards-in-time" (Lyngsø *et al.* 2005) and "along-the-genome" (Song and Hein 2003, 2005). Backwards-in-time approaches start with a data matrix and reduce it to an empty matrix through row and column operations corresponding to coalescence, mutation, and recombination events, which construct an ARG from the bottom up (Song *et al.* 2005; Wu 2008; Thao and Vinh 2019; Ignatieva *et al.* 2021). Along-the-genome approaches begin from an initial local tree at a single focal site. Moving the focal site along the genome changes the local tree via a subtree prune-and-regraft (SPR) operation whenever a recombination is encountered (Hein 1993; Wu 2011; Mirzaei and Wu 2017). Rasmussen and Guo (2023) focus on parsimonious fusion of local trees into an ARG, while the method described by Cámara *et al.* (2016) is based on topological data analysis. Reconstructing a parsimonious ARG for a given data set is NP-hard (Wang *et al.* 2001), so parsimony-based methods resort to heuristics and are limited to analyzing at most hundreds of sequences. Hence, a number of methods aim to balance computational efficiency with reconstruction of "reasonable," rather than parsimonious ARGs (Minichiello and Durbin 2006; Parida *et al.* 2008; Kelleher *et al.* 2019b; Speidel *et al.* 2019; Schaefer *et al.* 2021; Zhang *et al.* 2023).

An alternative approach is to treat the ARG as a latent parameter to be averaged out by Monte Carlo methods, based either on importance sampling (Griffiths and Marjoram 1996; Fearnhead

and Donnelly 2001; Jenkins and Griffiths 2011) or MCMC (Kuhner *et al.* 2000; Nielsen 2000; Kuhner 2006; Wang and Rannala 2008, 2009; O'Fallon 2013; Vaughan *et al.* 2017; Mahmoudi *et al.* 2022). These methods operate on representations of the "Little ARG" (see Appendix B), and are computationally expensive, being applicable to at most hundreds of samples consisting of tens or hundreds of kilobases with human-like parameters. State-of-the-art methods rely on cheaper, approximate models (Didelot *et al.* 2010; Heine *et al.* 2018; Hubisz and Siepel 2020; Hubisz *et al.* 2020; Medina-Aguayo *et al.* 2020; Deng *et al.* 2024). The most scalable method, SINGER, can be applied to hundreds of human genomes (Deng *et al.* 2024).

Methods to sample ARGs generate a "cloud" of estimates, and Kuhner and Yamato (2017) provide an approach to generate a set of consensus breakpoints and local trees from such a cloud. The approach is based on examining the recombination breakpoints in all of the input ARGS, and including those that are in at least $k$ of the input ARGs (with some additional filtering criteria) in the output. Within the resulting intervals, a consensus local tree is then generated using standard phylogenetic methods.

## Appendix D: Cell lineages and ARGs

In eukaryotes, ARGs are a result of the cellular processes of mitosis and meiosis. Mitosis leads to common ancestor events, and meiosis leads to recombination events (both crossover and gene conversion). Figure A2 shows a schematic of the events and the genomes (chromosome icons) that occur in the cellular germline of a simplified, diploid multicellular hermaphrodite eukaryote with partially overlapping generations. Here, an event is not represented by a specific genome. Rather, genomes can be associated with, or "tag", events above (ancestral to) or below (descended from) them. For example, tagging the 2 genomes above a recombination event leads to the 2 node representation seen in Figs. 1a and A1, whereas tagging the genome below a recombination event leads to the more conventional graphs in Figs. 3, 4a and b.

The schematic illustrates an important point about the biological reality of polytomies. Three lineages coalesce in the left-hand genome of individual $d_{10}$, but do so as the result of 2 successive bifurcations. This is necessarily so, because the only known method of reproducing DNA is by (semiconservative) duplication. Whether this polytomy is resolvable depends on the available mutational data. Mutations can occur along any cell lineages. For example, a mutation in the first cell division of $d_{10}$ could be shared between the 2 gametes produced by the cells in the left half of $d_{10}$ but not shared by the right-hand gamete. With enough mutations, each round of mitotic germline genome duplication within a single multicellular organism could in principle be distinguished.

## Appendix E: ARGs and local trees

The relationship between an ARG and its corresponding local trees is subtle and important. A fundamental property of genetics is that a given DNA nucleotide is inherited from exactly 1 parent genome, both at an organismal and cell-by-cell level (Appendix D). These paths of single-parent inheritance give rise, by definition, to a tree structure. As a result of recombination, adjacent nucleotides can have different paths of inheritance, and an ARG encodes the entire ensemble of local trees along the genome for a given set of sample nodes. Precisely defining the process by which local trees are extracted from an ARG is essential to our understanding of how ARGs and local trees are related, and we require a concrete mathematical structure to describe the local

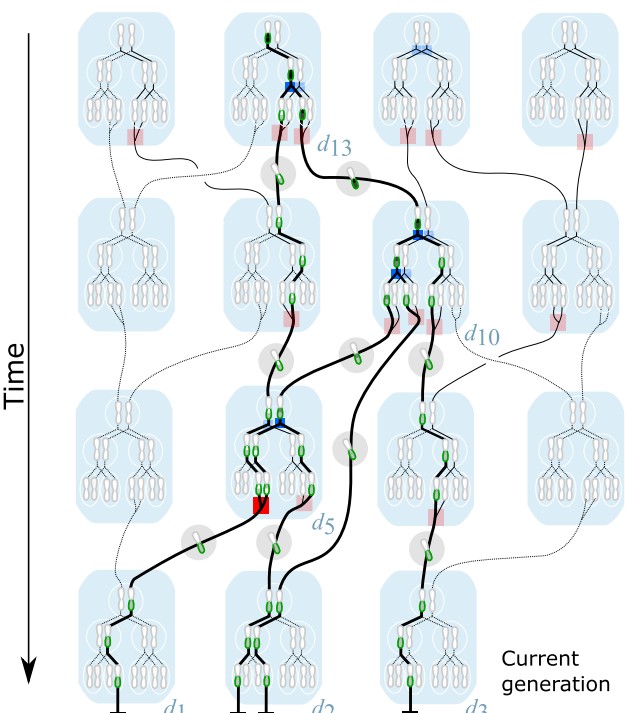

**Fig. A2.** Cellular inheritance of a single chromosome in a diploid population. Individuals (blue) contain diploid cells (white circles enclosing a homologous pair of chromosomes). For clarity, only 2 rounds of mitotic germ-line cell division are shown per individual, and meiosis is not illustrated in detail. Lines show prospective inheritance paths for all chromosomes. Solid lines show all possible retrospective ancestry paths for 4 chosen chromosomes (indicated by square black "sampling events") sampled from 3 diploid individuals ($d_1$, $d_2$, $d_3$) in the current generation. Ancestral recombination events and coalescence events are shown as red and blue squares, respectively. A realized ARG path for the lower arm of the sampled chromosomes is highlighted as a thick solid line, passing through a set of potential gARG nodes (green). This ARG involves a single recombination event and 4 coalescence events (highlighted as deep red and blue squares within individuals $d_5$, $d_{10}$, and $d_{13}$). ARG lineages also show gametic genomes, contained within shaded circles. As in Fig. 1a, inherited regions within the sampled chromosome arm are shaded by the number of descendant samples.

trees. It is important to note that although the following discussion is phrased in terms of the gARG encoding, the arguments apply equally to eARGs because any eARG can be converted to a gARG without loss of information (section *Ancestral material and sample resolution*).

Oriented trees provide a convenient formalism to capture these parent–child relationships in a well-defined combinatorial object. Let $\pi_1, \ldots, \pi_n$ be a sequence of integers, such that $\pi_u$ denotes the parent of node $u$, and $\pi_u = 0$ if $u$ is a root (Knuth 2011, p. 461). This encoding is particularly useful to describe evolutionary trees because parent–child relationships are important but the ordering of children at a node is not (Kelleher *et al.* 2013, 2014, 2016). Thus, for a given gARG with nodes $\{1, \ldots, n\}$ and edges $E$ (section *Genome ARGs*), we recover the local tree at position $x$ as follows. We begin by setting $\pi_u = 0$ for each $1 \leq u \leq n$. Then, for each sample node in $S$ we trace its path pastwards through the ARG for position $x$, and record this path in $\pi$. Specifically, at a given node $u$, we find an edge $(c, p, I) \in E$ such that $u = c$ and $x \in I$, and set $\pi_c \leftarrow p$. We then set $u \leftarrow p$, and repeat until either $\pi_u \neq 0$ (indicating we have traversed this section of the ARG already on the path from another sample) or there is no matching outbound edge (indicating we are at a root). Note that the local trees for an ARG are "sparse" (Kelleher

*et al.* 2016), because many ancestral nodes will not be reachable from the samples at a given position (so their corresponding entries in $\pi$ will be zero).

This combinatorial approach provides at least one novel insight, clarifying the fundamental relationship between ARGs and local trees. Suppose we are given a gARG defined by a set of nodes and edges. There is no requirement on the structure of this ARG beyond the basic definitions: it could correspond to an ARG in which every recombination event is exactly specified (e.g. Fig. 3) or one in which local trees are entirely disjoint (i.e. only the sample nodes are shared between them). If we are given the sequence of local trees for this gARG encoded as an oriented tree, along with the genome interval covered by each tree, we can recover the original gARG exactly. More formally, suppose we are given the local tree $\pi_1^x, \ldots, \pi_n^x$ for each nucleotide position $1 \le x \le L$ on a genome of length $L$. Then, the edges of the "local ARG" for this tree is given by $E^x = \{(u, \pi_u^x, \{x\}) \mid \pi_u^x \ne 0\}$. Because the ARG edges are defined by $(c, p, I)$ tuples, where the set $I$ defines the positions over which node $c$ inherits from parent $p$, we can then simply combine the "local ARGs" for each position $x$ to recover precisely the same set of edges as the original ARG. Thus, under this definition, there is a one-to-one correspondence between an ARG and the sequence of local trees that it encodes.

This is not the prevailing view, however. Kuhner and Yamato (2017) argue that the "interval-tree" representation of an ARG (the local trees and the genome intervals they cover) "does not contain all of the information in the underlying ARG: it lacks the number of recombinations occurring at each site, the times at which recombinations occurred, and the specific sequences involved as recombination partners." Shipilina *et al.* (2023) discuss the same ideas, and note that the "full ARG… contains more information than the series of tree sequences along the genome". These statements that an ARG contains more information than its local trees are true if we represent local trees in their conventional forms, but these forms discard important information that is available in an ARG.

There are 2 properties of how evolutionary trees are conventionally represented that lead to this disagreement about the relationship between local trees and an ARG. Firstly, the internal nodes of evolutionary trees are usually considered to be *unlabeled*, or equivalently, labeled by the leaves which they subtend. The same canonical labeling cannot be used for internal ARG nodes because the leaves they subtend will typically vary by genomic position. If we do not label the tree nodes in a way which is persistent across the sequence of local trees in the ARG, we lose the fact that the *same* ancestors sometimes persist across multiple trees. Defining ARG nodes as integers and using the oriented tree encoding explicitly labels internal nodes, and makes the relationship between tree and ARG nodes clear and precise.

The second property of how evolutionary trees are conventionally represented that is unhelpful in the context of ARGs is their focus on branching points (coalescences), i.e. nodes that have 2 or more children. As the introductory paragraph of this section emphasized, parent–child relationships are what fundamentally define a tree, and branching points can be seen as incidental. This is reflected by the oriented tree encoding which simply stores the local parent–child relationships, and does not, for example, directly tell us how many children a particular node has. The local tree at a given position records the *path* through the ARG; if this path omits nodes that are not branching points (such as $E$ in Fig. 1), information about the ARG is lost. We expand on this point in Appendices F and G where we which discuss "locally unary" nodes and the simplification process.

It is important to make the distinction here between the local trees that can be derived from a known ARG (as just discussed), and an ARG that can be derived from a sequence of *estimated* local trees. The ARG inference method Espalier (Rasmussen and Guo 2023) is illustrative in this context. It begins by splitting an input sequence alignment into segments that are assumed to be nonrecombining. Within each segment, an initial local tree is estimated using standard phylogenetic methods. By necessity, these local trees will contain internal nodes that are unlabeled and consist only of branching points: there is no information shared between the independent tree estimation steps across segments. Part of the task of stitching these trees together into an ARG is then, essentially, to generate labels for the internal nodes, and decide which nodes persist across multiple local trees. Espalier approaches this task by identifying maximal subtrees that do not change between pairs of adjacent local trees and then heuristically exploring the space of possible rearrangements of these subtrees. To derive details about recombination events, Espalier then attempts to infer the precise SPR operations (Hein 1990; Song 2003, 2006) induced by recombination between these partially reconciled local trees. Inferring the SPRs between leaf-labeled trees is NP-hard (Hein *et al.* 1996; Allen and Steel 2001; Bordewich and Semple 2005), but it is unclear what the complexity is when there is a degree of internal node sharing between trees. The combinatorial formulation of ARGs and local trees provided here may help clarify these fundamental questions.

## Appendix F: Locally unary nodes

As discussed in Appendix E, the local tree at a given position $x$ is best seen as the path through the ARG at that position, defined by the oriented tree $\pi_1^x, \ldots, \pi_n^x$. This path does not directly contain information about branching points, and defining a node's arity (number of child nodes) is therefore useful. The "local arity" of a node is the number of children it has in the local tree at position $x$, i.e. $a_u^x = |\{v : \pi_v^x = u\}|$ for each $1 \le u \le n$. The "ARG arity" of a node $u$ is the number of children it has in the graph topology, i.e. $a_u = |\{v : (v, u, I) \in E\}|$. Thus, the local arity is less than or equal to the ARG arity (more precisely, $0 \le a_u^x \le a_u$), and the local arity of a node may change as we move along the genome.

This distinction between ARG and local arity is mainly of interest when we consider nodes that have a single child: those that are *unary*. For the example in Fig. 1, nodes $G$ and $H$ are ARG-unary (Fig. 1b), and are consequently also unary in the local trees (Fig. 1c). On the other hand, node $F$ has 2 children in the graph, but is binary only in the local tree covering the interval $[2, 7)$, representing the coalescence of samples $A$ and $C$ in this genome region. Over the interval $[0, 2)$ no coalescence occurs, but we still record the fact that genome $C$ inherits from $F$ in the local tree. Thus, node $F$ has a single child in this interval: it is *locally unary*. In the same figure, $E$ is binary in the graph, being a common ancestor of $A$ and $C$, but is locally unary in all trees in which it is present. This is because no ancestral material coalesces in $E$: $A$ inherits genetic material from the far left-hand end of $E$, while $C$ only inherits the (disjoint) right-hand end.

By definition, ARG-unary nodes have 1 child but can have 1 or more parents. A node with 1 child and only 1 parent represents a "pass-through" node: these occur where we wish the record the passage of ancestral material through a known node. For example, in simulations it is sometimes useful to record the passage of ancestral material through known pedigree individuals regardless of whether common ancestry occurs. Nodes with 1 child and 2 parents arise when we model a recombination event using a single

node in the classical manner (e.g. Fig. 3). It is also possible for sample nodes to be ARG-unary, for example in inferences from longitudinal datasets where genetic data is sampled at many timepoints and recombination is rare, e.g. SARS-CoV-2 (Zhan *et al.* 2023).

More generally, locally unary nodes, which can have 1 or more children in the graph, are a common and important feature of many different types of ARG. As discussed in Appendix E, without these nodes marking the passage of ancestral material through specific ancestors, the local trees lack information about events other than local coalescence. For example, the local trees for the classical event ARG depicted in Fig. 2b follow the usual conventions and do not include any information about the recombination that occurred at node *D*. Given these 2 local trees in isolation we lack specific information about the recombination. Explicitly recording that node *D* lies on the branch joining *B* to *E* in the lefthand tree, and *B* to *F* in the right-hand tree resolves all ambiguity, and makes the collection of local trees exactly equivalent to the corresponding ARG. Unary nodes are a vital link between ARGs and local trees, and we cannot fully reason about how a local tree is embedded in an ARG without them.

## Appendix G:   Levels of simplification

ARG simplification is a powerful tool. In general, we can think of simplification as the process of removing nodes and rewriting edges (and their inheritance annotations) to remove various types of redundancy, much of which revolves around the presence of unary nodes (Appendix F). This successive removal of redundancy through a series of simplification steps is shown in Fig. 5.

The ARG in Fig. 5a is the output of a backwards-time Wright–Fisher simulation for a sample of 2 diploid individuals (population size $N = 10$), and follows a similar process to the methods described by Nelson *et al.* (2020). As we proceed backwards in time, generation by generation, the extant lineages choose parents randomly. With a certain probability recombination occurs, and the ancestral material of a lineage is split between the 2 parental genomes. Local coalescence occurs when lineages with overlapping ancestral material choose the same parent genome. Note that in this simulation we do not explicitly model recombination *events* via an ARG node, but simply record the *outcome* of a recombination via edges to the parent's 2 genomes. Thus, a recombinant node such as *G* in Fig. 5 may also correspond to a coalescence. The distinction of using a single node to represent a recombination event, as is done in Fig. 3, or 2 to represent the parent genomes, as in Fig. 5, is often not important. Either is possible in the gARG encoding, and the most convenient approach will vary by application. Note also that node *K* in Fig. 5 has 3 children. Polytomies like this are a natural feature of such a Wright–Fisher model. See Appendix D for a discussion of the ultimate biological interpretation of these topological considerations.

The graph visualizations in Fig. 5 have three novel features which require some explanation. Firstly, edge weights (the thickness of the lines joining nodes) correspond to the length of the inheritance intervals they are annotated with. This allows us to distinguish edges that persist across many local trees from those that are less influential (contrast the edge (*G*, *H*) with (*G*, *I*) in Fig. 5a). Secondly, node colors denote the number of parents that they have in the graph, allowing us to easily see roots (those with zero parents), recombinants (those with 2 parents) and more complex situations arising from simplification (see below). Thirdly, the shading intensity of a node denotes the "coalescent span," the fraction of the node's span (the length of genome in which it is

reachable from the samples in the local trees) over which it has more than 1 child. Nodes which are never locally unary, therefore, have a coalescent span of 100%, whereas nodes in which ancestral material never coalesces have a coalescent span of 0%.

Returning to the main topic of this section, Fig. 5a is the original simulation output, in which we retain all nodes involved in recombination or common ancestry events. This is the true history, and contains a very high level of detail, some of which may be considered redundant (or, from another perspective, unobservable). In Fig. 5a, the local trees (right) contain many unary nodes, fewer as we successively simplify (Fig. 5b and c), until we reach Fig. 5d, where there are none.

The first level of simplification that we can perform is based only on the graph topology. An example of graph topology that we may consider redundant (or nonidentifiable) is a "diamond" (Rasmussen *et al.* 2014) in which the 2 parent nodes of a recombination immediately join again into a common ancestor (e.g. *J*, *L*, *M* and *N* in Fig. 5a). Unless we are specifically interested in the recombination event or these ancestral genomes, the diamond can be replaced by a single edge without loss of information. More generally, any subgraph that is singly connected in both the leafward and rootward direction (a "super-diamond") can be replaced by 1 edge. This definition includes the case of a node that has 1 inbound and 1 outbound edge, such as nodes *F* and *H*. Figure 5b shows the result of this type of graph topology simplification.

Simplifying away diamonds will remove many unary nodes from the local trees, but there can still be nodes that are unary in all of the local trees. In particular, a node can represent a recombinant with multiple parents in the graph but only a single child (e.g. node *N* in Fig. 5b), or can represent a common ancestor with multiple children in the graph but in which no coalescence takes place in the local trees (node *R* in Fig. 5b). The distinction between the "common ancestry" of 2 or more genomes in an ancestral genome and the "coalescence" which may or may not occur in the local trees is important (Hudson 1983b; Kelleher *et al.* 2016). Consider *E* in Fig. 5a, for example. We can see from the graph that it is a common ancestor of samples *A* and *B*, but it does not correspond to any coalescence in the local trees to the left of position 44, and is, therefore, unary in these three trees. Such nodes are not singly connected in the graph, but are nevertheless unary in all of the local trees. The operation to remove them, therefore, requires knowledge not just of the graph topology but also of the ancestral material associated with the edges. As we see in Fig. 5c, removal of recombinant nodes can produce graph nodes with more than 2 parents (e.g. node *E*); and likewise, removal of common ancestor but noncoalescent nodes can produce graph nodes with more than 2 children (e.g. node *S*). Both cases represent the merged *effects* of multiple evolutionary events in a single node (genome), and the ARG no longer contains the intermediate genomes corresponding to those events (see also Appendix D).

The remaining nodes are MRCAs of some subset of the samples at *some* positions along the genome. We still have some unary nodes in the local trees, but these nodes will correspond to a coalescence in at least 1 other local tree. For example, node *K* is unary in the 4th tree of Fig. 5c, but is either binary or ternary in all other local trees (recall this is a Wright–Fisher simulation). The final level of simplification is to alter the edge annotations such that, although no nodes are removed from the graph, all unary nodes disappear from the local trees (Fig. 5d). Note that although this last stage produces simpler local trees, by removing information about the exact paths taken by lineages through the graph, we lose potentially useful information about shared edges between trees. The msprime simulator, and the version of

Hudson's algorithm described by Kelleher *et al.* (2016), produce ARGs that are fully simplified (i.e. contain no locally unary nodes). It is not difficult, however, to update these methods to record information about the passage of ancestral material through genomes under a range of conditions.

## Appendix H: Precision of recombination information

As illustrated in Fig. 5, successive levels of ARG simplification reduce the amount of information about the history of the sample that is stored. Some of the information lost, e.g. "diamond" removal (Fig. 5a), seems like a reasonable tradeoff for a simpler structure. The consequences of other simplifications, however, are more subtle and relate directly to what can be known about recombination events and the levels of precision that we should seek to infer about them.

The ARGs in Fig. 5 contain different numbers of local trees (6, 5, 5, and 4, respectively, for a–d). When we move from a to b the local trees for the intervals [44, 61) and [61, 87) are merged because the only differences between them are their paths through nodes *L* and *M*. These nodes that participated in the diamond are removed from the ARG, and we have lost all information about the corresponding recombination at position 61. Other nodes (e.g. *O* and *P*) have also been removed but these represent the *parents* of recombinants. The recombinant nodes themselves (e.g. *N*) are still present, and represent precise information about the time, genomic location and lineages involved in the recombination event.

Figure 5c has the same number of local trees as Fig. 5b, but has less precise information about recombination. Continuing the previous example, node *N* has been removed from the graph because it was unary in all of the local trees; its outbound edges to *S* and *Q* have effectively been "pushed down" to *E* (which is retained because it is the coalescent parent of *A* and *B* over the interval [44, 100)). We have, therefore, lost precision about the *timing* of this recombination event, and know only that it must have occurred between the times of node *E* and *Q*.

Figure 5d removes all unary nodes from the local trees, and further reduces the precision of recombination information. Node *E* has not been removed from the graph because it is coalescent in the final tree, but we no longer know that the recombination event at position 30 was ancestral to it, or have any indication of its timings. Furthermore, trees for [44, 87) and [87, 100) were only distinguishable by the passage of the former tree through nodes *E* and *Q*, and so the recombination on node *G* at position 87 has been lost entirely.

## Appendix I: Example inferred ARGs

The scalability gains made by recent ARG inference methods such as `Relate` (Speidel *et al.* 2019) and `tsinfer` (Kelleher *et al.* 2019b) have been, in part, due to inferring lower levels of precision about recombination than classical methods. Neither method infers explicit recombination events, and therefore their outputs cannot be described using the classical eARG formalisms (section *Event ARGs*). Nonetheless, both methods produce estimates in which nodes and edges persist across multiple trees, creating inheritance graphs which fit naturally into the gARG formulation. To illustrate the varying levels of information captured by current methods, and some qualitative differences between them, Fig. 4 shows graphical depictions of example ARGs produced by 4 tools using substantially different inference strategies.

The first 2 methods explicitly infer recombination events. `KwARG` (Ignatieva *et al.* 2021) is a parsimony-based approach which searches the space of plausible ARGs, outputting minimal ones using heuristics. `ARGweaver` (Rasmussen *et al.* 2014) on the other hand is model-based, sampling from a discretized version of the SMC (McVean and Cardin 2005; Marjoram and Wall 2006). Note that both `KwARG` and `ARGweaver` produce many ARGs, and those shown in Fig. 4 are arbitrarily selected examples. While the second 2 methods both produce a single best-guess estimate and do not explicitly infer recombination events, they are based on quite different principles. `Tsinfer` works in a 2-step process, first generating ancestral haplotypes via heuristics and then inferring inheritance relationships between them using the Li and Stephens model (Li and Stephens 2003). `Relate` first reconstructs local tree topologies across the genome, using a variant of the Li and Stephens model to estimate the ordering of coalescence events in each tree, and then estimates branch lengths using MCMC with a coalescent-based prior. See Appendix C for more details on these and other inference methods.

Inferred ARGs are based on the Kreitman (1983) dataset, a standard benchmark in the classical ARG literature. It consists of 43 biallelic SNPs spanning 2.4 kb of the *D. melanogaster* ADH locus on chromosome 2L. Where required for inference purposes we assume mutation and recombination rates of $5.49 \times 10^{-9}$ and $2.40463 \times 10^{-9}$ per site per generation (Comeron *et al.* 2012; Schrider *et al.* 2013) and a constant effective population size of 1,720,600 (Li and Stephan 2006), as provided by the `stdpopsim` catalog (Adrion *et al.* 2020; Lauterbur *et al.* 2023). Software versions were `KwARG` v1.0, `ARGweaver-D` (2019), `tsinfer` v0.3.1, and `Relate` v1.1.9. Full details and code for generating these figures are available on GitHub (see *Data availability*).

Considering Fig. 4, we can see that there is substantial variation in the number of recombination breakpoints inferred by different methods, with e.g. `ARGweaver` suggesting far more than the 7 required for this dataset under minimal parsimony assumptions (Song and Hein 2003). A sense of the amount of recombination in each ARG is provided by the node coloring scheme, which shows the number of parents for each node. In Fig. 4a and b, each recombination event corresponds to a node with exactly 2 parents and 1 child. As these methods explicitly infer a recombination event for each breakpoint, the number of breakpoints equals the number of 2-parent (brown) nodes. In contrast, Fig. 4c and d do not have this straightforward relationship between the number of nodes with multiple parents and number of breakpoints along the genome. In both ARGs the number of breakpoints is smaller than the number of multiple-parent ARG nodes, showing that several multiple-parent nodes must share breakpoint positions. There are also ARG nodes with multiple parents and multiple children, where 1 or more recombinations have been pushed down onto a more recent node. As a consequence, it may be difficult to condense each transition between trees in these ARGs into a set of SPR operations.

Shading within nodes in Fig. 4 indicates the fraction of the node's span over which it is coalescent (Appendix F). For example, brown nodes in Fig. 4a and b are clear because there is no local coalescence at these recombination nodes (they are "ARG-unary," and so local coalescence is impossible). The significant number of partially shaded nodes in Fig. 4a–c demonstrates that the `KwARG`, `ARGweaver`, and `tsinfer` ARGs all contain locally unary nodes. Another difference between methods highlighted in this figure is the presence of polytomies, which only `tsinfer` creates. The most obvious example involves nodes `Fr-F`, `Wa-F`, and `Af-F`, which happen to have identical sequences. Because `KwARG`,

ARGweaver, and Relate require bifurcating trees by design, each picks an arbitrary order of branching (hence Fig. 4a and b disagree in this order, and Fig. 4d even shows different orders in different trees).

The bottom row of Fig. 4 shows the extent to which graph edges persist along the genome. All 4 methods infer nodes and edges that are shared between multiple trees, to varying degrees. For example, all of the methods infer that Af-f, Fr-f, and Wa-f form a clade along the entire sequence. In particular, we can see both

tsinfer and (to a lesser extent) Relate have edges that span multiple tree boundaries, indicating that they are not inferring a series of *unrelated* local trees. However, in comparison to KwARG and ARGweaver neither method results in extensive node sharing in the oldest time periods. Overall, Fig. 4 shows that tsinfer and Relate ARGs contain a level of detail that lies somewhere between a sequence of unrelated local trees on one extreme and an ARG with precisely specified recombination events on the other (Fig. 4a and b).