## [Peer Review File · Genetics]

A general and efficient representation of ancestral recombination graphs

Yan Wong, Anastasia Ignatieva, Jere Koskela, Gregor Gorjanc, Anthony Wohns, and Jerome Kelleher

NOTE: The reviews and decision letters are unedited and appear as submitted by the reviewers.

In extremely rare instances and as determined by a Senior Editor or the EIC, portions of a review may be redacted. If a review is signed, the reviewer has agreed to no longer remain anonymous.

The review history appears in chronological order.

Review Timeline:

Submission Date:	2023-11-03
Editorial Decision:	2024-02-08
Resubmission Received:	2024-04-22
Editorial Decision:	2024-05-31
Resubmission Received:	2024-06-04
Accepted:	2024-06-05

February 5, 2024

GENETICS-2023-306610

A general and efficient representation of ancestral recombination graphs

Dear Dr. Kelleher:

Below you will find reviews from a reviewer and myself. One of our original reviewers accepted and the dropped completely out of contact over the winter holidays. Thus I took on writing the review myself. I apologize for my slow turn around time, January was a very busy time for me with teaching and family commitments.

There's a lot of useful insights in this piece and view it as a useful contribution. However, both the reviewer and I wondered who the audience was for this piece and felt like the format did not really suit the goals of the article. While your manuscript is not currently acceptable for publication in GENETICS, we would welcome a very heavily revised manuscript. My main suggestion is to work out who your average reader is and then simplify and shorten the article significantly with them in mind. If you are positioning this as the place to go and find out what a genome ARG is you want it to be far more accessible. Both reviewers have comments and concerns to be fully addressed in a revised manuscript. You can read their reviews at the end of this email.

We look forward to receiving your revised manuscript. Please let the editorial office know approximately how long you expect to need for revisions.

Upon resubmission, please include:

1. A clean version of your manuscript;
2. A marked version of your manuscript in which you highlight significant revisions carried out in response to the major points raised by the editor/reviewers (track changes is acceptable if preferred);
3. A detailed response to the editor's/reviewers' feedback and to the concerns listed above. Please reference line numbers in this response to aid the editor and reviewers.

Your paper will likely be sent back out for review.

Additionally, please ensure that your resubmission is formatted for GENETICS
<https://academic.oup.com/genetics/pages/general-instructions>

Follow this link to submit the revised manuscript: Link Not Available

Sincerely,

Graham Coop
Associate Editor
GENETICS

Approved by:
Hongyu Zhao
Senior Editor
GENETICS

Associate Editor's overview & review

I gained a number of insights from reading the paper, however, I wonder who the intended audience of the paper is. It is not a research paper, indeed many of the technical details are given in other papers. Nor is it written at a particularly accessible level for people trying to understand ARGs and tree sequences. The paper is advocating for the adoption of "genome ARG (gARG)" and its simplified versions, but the authors need to be much clearer who they want to adopt this and in what level. Separating our structure for storing large ARGs estimated from data, from the "true" ARG seems like a very useful argument to make but this gets lost a lot in the details.

The "event ARG" is hard for people to grapple with on first brush, however, in my view, the authors' presentation here risks making ARGs even less intuitive. If the paper is aimed at convincing a broad audience to adopt this framework, it needs to be simplified and written in a much clearer and more accessible way.

My view is that the authors should switch the article to being a "Perspective" or "Review" and shorten it by moving a lot of the technical detail to the appendices. I suspect this switch will help people understand that this is the author's take on a topic, and

also help it be read through by a far broader audience. People interested in the details can then work through the appendices without getting lost in them on the first read through. I would be interested to see such an article published in Genetics, as I think it would be an important contribution and help clarify a complex field. I'm happy for the authors to argue for keeping different sections in the main paper, but in doing so they need to be more clearly justified to the reader in the paper.

The authors make a strong case that the data does not contain information about certain aspects of the ARG, that does not mean that probabilistic statements about these events cannot be made under particular models (or that these models cannot be learnt from data). Coalescent inference often involves integrating out events that we cannot observe in the data and dealing with great uncertainty in the timing of events. I think the authors should perhaps be more open to the view that for some applications we may want to build back in more events even if they are not "needed" in some minimum construction. At the moment it reads a little dogmatically, when I suspect that is not the authors true position.

It seems in general that many in the field, at least methods-savvy folks, are on board with tree sequences representation of data. If the "succinct tree sequence data structure (usually known as a "tree sequence" for brevity) is a practical gARG implementation", why not just leave "tree sequence" as the term we use. It is far less clear to me that we need to label yet another object an ARG, without muddying the definition of an ARG even further. More justification is needed to help the reader understand what the authors are arguing here. The authors could also do with having a clearer separation of the "gARG" from the "fully simplified gARG" as for a lot of the paper it feels like they informally move back and forth of arguing the strengths of both without being fully clear on which they are talking about.

More detailed comments:

Introduction

The authors state that concepts : "that we must be careful to distinguish are the "true" ARG, describing the actual history of a sample of genomes, and a "population" ARG which is the true ARG of every individual in a population.". But given that they're both "true" ARGs, one for a sample and one for the whole population sample it's not clear what is that different here. Also from the description I think the authors mean the population ARG to not include every individual who has ever lived, but it's not clear.

"population-scale true ARGs unquestionably exist, they can also never be entirely known," that's true of all sample ARGs

"perfectly resolved into binary splits and mergers" - not clear which direction time is running in this statement.

Subsequent sections:

"For example, gene conversion could be accommodated either by stipulating a third type of event (annotated by two breakpoints and corresponding traversal conventions for recovering the local trees)" -Cite the coalescent with gene conversion papers

"we will refer to this Griffiths encoding as the "event ARG" - It feels a little strange to attach a single person's name to an approach that the paper takes a fairly negative take on. I understand there's no ill feeling here, but it's not clear how well this comes across.

Briefly define "Big ARG" and "Little ARG" in the main text as they are referred to multiple times.

Section 4 and Figure 3. - There's a lot of detail in this section to then get to this sentence at the end: "Note that the Wiuf and Hein (1999b) eARG in Fig. 3 is not particularly representative, because inference or simulation methods usually only generate ARGs containing nodes and edges ancestral to the sample". So this feels like the authors have attached a lot of baggage to the eARG that it didn't actually have.

Sections 5 & 6 similarly are fairly technical and it's not fully clear who they are aimed at, Could these sections be moved to the appendix?

Section 7 and 8: Again there is a lot of detail here, much of which amounts to a long description of Figure 4. It is not fully clear if the authors want the bottom graph in Figure 4 to be the end product (a "fully simplified gARG"), if that is the case they need to state this up front and label it that way in the graph. As I mention above, it's not really clear to me that labeling 4D an ARG is really useful, given how much has been stripped away.

Section 9: I'm not sure what the goal of this section is.

Discussion:

"Fully decoupling the general concept of an ARG from the coalescent with recombination (hence-forth, "coalescent") is an important step." Not sure it matters whether the coalescent is a good or bad model for data. What matters is the separation of the idea of genetic genealogy from stochastic models that generate such structures. That separation occurred long before this paper, and I'm not sure that discussing the limitations of the stochastic model adds much to what is already a long paper. More generally, please work through the discussion with the aim of

significantly shortening it.

Reviewer #1 (Comments for the Authors (Required)):

In this article, Wong et al. describe a data structure called the "genome ARG" (gARG), designed to effectively represent information in ancestral recombination graphs. The authors have done an excellent job of clearly explaining the concept and offering a detailed comparison of different ARG definitions. The practicality of the gARG encoding is already well recognized in the community through various applications, such as msprime, forward-time simulation, and the efficient computation of population genetic statistics.

However, I have reservations about categorizing this work as a standalone "Investigation" article. It seems to align more closely with the nature of a review article. While the authors claim to be "proposing an alternative mathematical formulation," this characterization might be somewhat of an overstatement. The contribution, while valuable, appears to be more in the realm of summarizing and clarifying existing concepts rather than introducing a fundamentally new mathematical approach.

Minor comments:

The figures provided are generally helpful; however, the font size in some of them is quite small, making them difficult to read. For instance, the node labels in Figure 3C are impossible to read without zooming in maximally. I suggest increasing the font size in most figures to ensure clear legibility.

Associate Editor Comments:

Response to the editor

Thank you for considering this manuscript for publication. We have addressed the points you and Reviewer 1 raised below in turn.

Associate Editor's comments

Point 1.1 — I gained a number of insights from reading the paper, however, I wonder who the intended audience of the paper is. It is not a research paper, indeed many of the technical details are given in other papers. Nor is it written at a particularly accessible level for people trying to understand ARGs and tree sequences. The paper is advocating for the adoption of “genome ARG (gARG)” and its simplified versions, but the authors need to be much clearer who they want to adopt this and in what level. Separating our structure for storing large ARGs estimated from data, from the “true” ARG seems like a very useful argument to make but this gets lost a lot in the details.

Reply: We have clarified the purpose of the article by explicitly identifying “ARG practitioners” as the intended audience. We hope that it may serve to clarify some points of confusion among this community, and provide a useful starting point into a large and confusing literature for those new to the field.

Point 1.2 — The “event ARG” is hard for people to grapple with on first brush, however, in my view, the authors’ presentation here risks making ARGs even less intuitive.

Reply: We have attempted to simplify this by cutting out extraneous detail and focusing on the most important points that distinguish an Event ARG from a Genome ARG.

Point 1.3 — My view is that the authors should switch the article to being a “Perspective” or “Review” and shorten it by moving a lot of the technical detail to the appendices. I suspect this switch will help people understand that this is the author’s take on a topic, and also help it be read through by a far broader audience. People interested in the details can then work through the appendices without getting lost in them on the first read through. I would be interested to see such an article published in *Genetics*, as I think it would be an important contribution and help clarify a complex field. I’m happy for the authors to argue for keeping different sections in the main paper, but in doing so they need to be more clearly justified to the reader in the paper.

Reply: We agree that the article is much more suited to the “Perspective” format, and have heavily revised it following your suggestions. We have kept only what we regard as sections essential to the main messages of the paper in the main text. Briefly, we clearly need the Genome ARGs and Event ARGs sections, and the “Ancestral material and sample resolution” section is critical to understanding how they relate to each other. We have added a new “A Diversity of Structures” section, which highlights some important nuances possible when using gARGs as illustrated by Figures 4 and 5. We regard Section 6 as vital, as it explicitly draws the connection between gARGs, eARGs and the tskit library.

Point 1.4 — The authors make a strong case that the data does not contain information about certain aspects of the ARG, that does not mean that probabilistic statements about these events

cannot be made under particular models (or that these models cannot be learnt from data). Coalescent inference often involves integrating out events that we cannot observe in the data and dealing with great uncertainty in the timing of events. I think the authors should perhaps be more open to the view that for some applications we may want to build back in more events even if they are not “needed” in some minimum construction. At the moment it reads a little dogmatically, when I suspect that is not the authors true position.

Reply: We have cut this text.

Point 1.5 — It seems in general that many in the field, at least methods-savvy folks, are on board with tree sequences representation of data. If the “succinct tree sequence data structure (usually known as a “tree sequence” for brevity) is a practical gARG implementation”, why not just leave “tree sequence” as the term we use. It is far less clear to me that we need to label yet another object an ARG, without muddying the definition of an ARG even further. More justification is needed to help the reader understand what the authors are arguing here.

Reply: This was our initial position, but it has become clear that “tree sequence” is regularly confused with “a sequence of disconnected trees” and there are numerous inaccurate statements along these lines in the literature. There was a deep-seated confusion about the relationship between what is classically considered an ARG, and the structures that Relate, tsinfer and ARG-Needle produce. We feel strongly that the only way to resolve this is to show that “tree sequences” *are* ARGs, and that there needs to be *some* terminology to distinguish the classical “full” ARG that is driven by events, and the more general structure used by tskit. Thus, in order to tackle this “tree sequences aren’t proper ARGs” sentiment, the terminology must contain “ARG”. Our hope is that the term Genome ARG will not be necessary in the future, and we will simply say “ARG”.

We hope that the shorter main text will make the requirement for a simple general definition of an ARG data structure clear, without needing to explicitly mention these concerns in the text.

Point 1.6 — The authors could also do with having a clearer separation of the “gARG” from the “fully simplified gARG” as for a lot of the paper it feels like they informally move back and forth of arguing the strengths of both without being fully clear on which they are talking about.

Reply: We have clarified this point in several places by stating when we assume a fully simplified ARG and not.

Point 1.7 — (Introduction) The authors state that concepts : “that we must be careful to distinguish are the “true” ARG, describing the actual history of a sample of genomes, and a “population” ARG which is the true ARG of every individual in a population.”. But given that they’re both “true” ARGs, one for a sample and one for the whole population sample it’s not clear what is that different here. Also from the description I think the authors mean the population ARG to not include every individual who has ever lived, but it’s not clear.

“population-scale true ARGs unquestionably exist, they can also never be entirely known,” that’s true of all sample ARGs

“perfectly resolved into binary splits and mergers” - not clear which direction time is running in this statement.

Reply: We have cut these passages.

Point 1.8 — “For example, gene conversion could be accommodated either by stipulating a third type of event (annotated by two breakpoints and corresponding traversal conventions for recovering the local trees)” -Cite the coalescent with gene conversion papers

Reply: Thanks for pointing out this omission. We have cited Wiuf and Hein, 2000, but are not aware of other coalescent with gene conversion papers. We would be happy to cite others if you provide references.

Point 1.9 — “we will refer to this Griffiths encoding as the “event ARG” - It feels a little strange to attach a single person’s name to an approach that the paper takes a fairly negative take on. I understand there’s no ill feeling here, but it’s not clear how well this comes across.

Reply: Thanks for pointing this out — you are absolutely right. We only mention Griffiths in a few places now, and avoid attaching his name to the structure.

Point 1.10 — Briefly define “Big ARG” and “Little ARG” in the main text as they are referred to multiple times.

Reply: They are no longer mentioned in the main text.

Point 1.11 — Section 4 and Figure 3. - There’s a lot of detail in this section to then get to this sentence at the end: “Note that the Wiuf and Hein (1999b) eARG in Fig. 3 is not particularly representative, because inference or simulation methods usually only generate ARGs containing nodes and edges ancestral to the sample”. So this feels like the authors have attached a lot of baggage to the eARG that it didn’t actually have.

Reply: We have cut this paragraph.

Point 1.12 — Sections 5 & 6 similarly are fairly technical and it’s not fully clear who they are aimed at, Could these sections be moved to the appendix?

Reply: This has been done.

Point 1.13 — Section 7 and 8: Again there is a lot of detail here, much of which amounts to a long description of Figure 4.

Reply: We have moved these to appendices.

Point 1.14 — It is not fully clear if the authors want the bottom graph in Figure 4 to be the end product (a “fully simplified gARG”), if that is the case they need to state this up front and label it that way in the graph.

Reply: We have added a clarification to the caption to say that this is ‘often described as “fully simplified”’.

Point 1.15 — As I mention above, it’s not really clear to me that labeling 4D an ARG is really useful, given how much has been stripped away.

Reply: On this point we disagree. As discussed above and argued in the text, we feel that it is imperative that there is a simple formal definition of an ARG that corresponds to informal usage. Having some arbitrary threshold of “sufficient structure” for something to be considered a ‘proper’ ARG can surely only lead to more confusion. The only reasonable path forward is for the term ARG to become general, and for the community to invent adjectives to classify the different types of ARG. As we point out in the introduction, structures like this (the default output of msprime) are informally defined as ARGs by emerging consensus, and so the key point of this manuscript is to provide a corresponding formal definition.

Point 1.16 — Section 9: I’m not sure what the goal of this section is.

Reply: This section has been moved to an appendix. It is intended to clarify the common features of existing ARG inference methods, as well as highlight their differences.

Point 1.17 — Discussion: “Fully decoupling the general concept of an ARG from the coalescent with recombination (hence-forth, “coalescent”) is an important step.” Not sure it matters whether the coalescent is a good or bad model for data. What matters is the separation of the idea of genetic genealogy from stochastic models that generate such structures. That separation occurred long before this paper, and I’m not sure that discussing the limitations of the stochastic model adds much to what is already a long paper. More generally, please work through the discussion with the aim of significantly shortening it.

Reply: We have cut this passage and several others, substantially shortening the discussion.

Reviewer 1

Point 2.1 — In this article, Wong et al. describe a data structure called the “genome ARG” (gARG), designed to effectively represent information in ancestral recombination graphs. The authors have done an excellent job of clearly explaining the concept and offering a detailed comparison of different ARG definitions. The practicality of the gARG encoding is already well recognized in the community through various applications, such as msprime, forward-time simulation, and the efficient computation of population genetic statistics.

Reply: Thank you for the kind words, we are glad you find the exposition clear.

Point 2.2 — However, I have reservations about categorizing this work as a standalone “Investigation” article. It seems to align more closely with the nature of a review article. While the authors claim to be “proposing an alternative mathematical formulation,” this characterization might be somewhat of an overstatement. The contribution, while valuable, appears to be more in the realm of summarizing and clarifying existing concepts rather than introducing a fundamentally new mathematical approach.

Reply: We are absolutely in agreement, and have restructured the article as a “Perspective”.

Point 2.3 — The figures provided are generally helpful; however, the font size in some of them is quite small, making them difficult to read. For instance, the node labels in Figure 3C are impossible

1st Revision - Authors' Response to Reviewers: April 22, 2024

to read without zooming in maximally. I suggest increasing the font size in most figures to ensure clear legibility.

Reply: We have modified Figure 3 to make it more legible, and will endeavour to adapt the figures to be legible in the final published typesetting (should it be accepted!).

May 31, 2024

GENETICS-2024-307040

A general and efficient representation of ancestral recombination graphs

Dear Dr. Kelleher:

I am pleased to inform you that, with minor revisions, it is potentially suitable for publication in GENETICS. My comments and concerns that need to be addressed in a revised manuscript are at the end of this email.

We look forward to receiving your revised manuscript. Please let the editorial office know approximately how long you expect to need for revisions.

Upon resubmission, please include:

1. A clean version of your manuscript;
2. A marked version of your manuscript in which you highlight significant revisions carried out in response to the major points raised by the editor/reviewers (track changes is acceptable if preferred);
3. A detailed response to the editor's/reviewers' comments and to the concerns listed above. Please reference line numbers in this response to aid the editors.

Additionally, please ensure that your resubmission is formatted for GENETICS.

<https://academic.oup.com/genetics/pages/general-instructions>

Follow this link to submit the revised manuscript: Link Not Available

Sincerely,

Graham Coop
Associate Editor
GENETICS

Approved by:
Hongyu Zhao
Senior Editor
GENETICS

Associate Editor Comments:

The paper is much improved and the main points come through much more clearly. I thank the authors for all their work on these revisions. I've listed some minor comments below but I'm happy for the authors to treat these comments as minor revisions and choose how/if they resolve them. I certainly have gained a lot by reading through this and think it's a useful addition to the field.

My remaining broad concern is that the paper is still in places somewhat narrow about the goals of future ARG development. I certainly see the practical utility of dropping inference down to some minimum "knowable" structure that can be reconstructed using deterministic algorithms for very large datasets. However, probabilistic reconstructions of some form of ARG with more explicit events is also a reasonable goal moving forwards (e.g. for some applications we may want a subset of the recombination events explicitly included). There are a few places where the paper still comes across as overly dogmatic about the minimum "knowable" ARG being the only goal (although the discussion casts a broader view).

Abstract: "This approach is out of step with modern developments, which do not represent genetic inheritance in terms of these events or explicitly infer them." So this is on the places where I feel like the authors state things too strongly. The authors, and some others, approaches have taken this path, but folks can agree that the gARG is a good idea and yet think that explicitly inferring details of recombination events is a 'modern' goal.

"Broadly speaking, an ARG describes the different paths of genetic inheritance caused by recombination, encapsulating the resulting complex web of genetic ancestry" - add "of a set of samples". Also I'd say "genetic ancestors", as ancestry is tied up

with genetic ancestry groups in peoples' minds.

"We define a genome as the complete set of genetic material that a child inherits from one parent. A diploid individual therefore carries two genomes, one inherited from each parent (we assume diploids here for clarity, but the definitions apply to organisms of arbitrary ploidy). " -Excludes Y, mtDNA, and X as written, please revise, e.g. talk about autosomal genome.

"The topology of a gARG specifies that genetic inheritance occurred between particular ancestors and descendants, " -struggle slightly with word "particular" here as the identity of the ancestors is not known. Deleting "particular" is likely sufficient. "This is sufficient to describe the effects of inheritance under any form of homologous recombination (such as multiple crossovers,..." -do you mean multiple crossovers during a single round of meiosis.

"In this encoding there are two types of internal node in the graph, representing the common ancestor and recombination events in the history of a sample. " stipulate that these are most recent common ancestor events.

"This approach assumes all events are knowable, and does not provide an obvious mechanism for either aggregating multiple events or expressing uncertainty about them. While this is not a problem when describing the results of simulations". -Maybe one way to flip this around would be to say that because it arose from tracking a particular stochastic process it has these properties. Also I don't think it assumes that all events are knowable, eg we could construct some parsimonious ARG or probabilistic ARG. If we wish to express uncertainty about events we usually give draws from the posterior etc. I agree that might be computational prohibitive with large samples etc, but it seems like place to take a broad view. This seems like a place to acknowledge that for some applications we might want to explicitly reconstruct the events.

"A key feature of the gARG encoding is that it enables these varying levels of precision to be represented, and brings these nuanced features to light." -the word nuanced feels strange here.

"Simpler representations can be formed by removing "unknowable" nodes (Fig. 5B)" -unknowable is vague here, do you mean bubbles along a single lineage?

"The gARG encoding leads to highly efficient storage and processing of ARG data, "-As gARG has various levels of precision, perhaps this needs to state that the "gARG encoding can lead to..." or be more precise that this is a reduced precision level.

"The succinct tree sequence data structure (usually known as a "tree sequence" for brevity) is a practical gARG implementation focused on efficiency." - If the tree sequence is focused at a particular level of gARG simplification be precise about this.

"Methods targeting large-scale datasets tend to simplify the inference problem by making a single, deterministic best-guess " --I think this is the best guess of the topology, and the uncertainty in times given the ARG is downstream of this. If so please clarify. Also I'd perhaps explicitly acknowledge Deng et al (SINGER), e.g. "deterministic best-guess of the topology (see Deng et al for parallel developments addressing uncertainty with somewhat small sample sizes)" or something like that. While these deterministic approaches are a strong way forward for human biobank scale data, it's good to be highlight parallel developments that might be key to other applications.

Response to the editor

We thank you again for a careful reading of this manuscript, and the further suggestions for improving its clarity and flow. We have addressed the points you raised below in turn.

Associate Editor's comments

Point 1.1 — My remaining broad concern is that the paper is still in places somewhat narrow about the goals of future ARG development. I certainly see the practical utility of dropping inference down to some minimum “knowable” structure that can be reconstructed using deterministic algorithms for very large datasets. However, probabilistic reconstructions of some form of ARG with more explicit events is also a reasonable goal moving forwards (e.g. for some applications we may want a subset of the recombination events explicitly included). There are a few places where the paper still comes across as overly dogmatic about the minimum “knowable” ARG being the only goal (although the discussion casts a broader view).

Reply: We have gone through the article and, in addition to the suggestions made below, have rephrased parts to make it clear that a gARG can be used to encode a *variety* of ARG structures, whether events are or are not explicitly inferred by the reconstruction method. We specifically state at the end of *A diversity of structures* that

A gARG can encode a diversity of ARG structures, including those where events *are* recorded explicitly, and those where they are treated as fundamentally uncertain and thus not explicitly inferred (Appendix H).

Point 1.2 — Abstract: “This approach is out of step with modern developments, which do not represent genetic inheritance in terms of these events or explicitly infer them.” So this is on the places where I feel like the authors state things too strongly. The authors, and some others, approaches have taken this path, but folks can agree that the gARG is a good idea and yet think that explicitly inferring details of recombination events is a ‘modern’ goal.

Reply: We have changed this to “This approach is out of step with some modern developments, however,…”

Point 1.3 — “Broadly speaking, an ARG describes the different paths of genetic inheritance caused by recombination, encapsulating the resulting complex web of genetic ancestry” - add “of a set of samples”. Also I’d say “genetic ancestors”, as ancestry is tied up with genetic ancestry groups in peoples’ minds.

Reply: Amended as suggested.

Point 1.4 — “We define a genome as the complete set of genetic material that a child inherits from one parent. A diploid individual therefore carries two genomes, one inherited from each parent (we assume diploids here for clarity, but the definitions apply to organisms of arbitrary ploidy). ” -Excludes Y, mtDNA, and X as written, please revise, e.g. talk about autosomal genome.

Reply: Amended as suggested.

Point 1.5 — “The topology of a gARG specifies that genetic inheritance occurred between particular ancestors and descendants,” -struggle slightly with word “particular” here as the identity of the ancestors is not known. Deleting “particular” is likely sufficient.

Reply: Amended as suggested.

Point 1.6 — “This is sufficient to describe the effects of inheritance under any form of homologous recombination (such as multiple crossovers,...” -do you mean multiple crossovers during a single round of meiosis.

Reply: Yes - amended to clarify this.

Point 1.7 — “In this encoding there are two types of internal node in the graph, representing the common ancestor and recombination events in the history of a sample. ” stipulate that these are most recent common ancestor events.

Reply: Amended as suggested.

Point 1.8 — “This approach assumes all events are knowable, and does not provide an obvious mechanism for either aggregating multiple events or expressing uncertainty about them. While this is not a problem when describing the results of simulations”. -Maybe one way to flip this around would be to say that because it arose from tracking a particular stochastic process it has these properties. Also I don't think it assumes that all events are knowable, eg we could construct some parsimonious ARG or probabilistic ARG. If we wish to express uncertainty about events we usually give draws from the posterior etc. I agree that might be computational prohibitive with large samples etc, but it seems like place to take a broad view. This seems like a place to acknowledge that for some applications we might want to explicitly reconstruct the events.

Reply: We have rephrased this part to read

This approach requires all events to be recorded explicitly, and does not provide an obvious mechanism for aggregating multiple, potentially unresolvable, events. As datasets approach the population scale [citations] representing such uncertainty directly through the data structure is a useful alternative to classical methods based on probabilistic sampling.

Point 1.9 — “A key feature of the gARG encoding is that it enables these varying levels of precision to be represented, and brings these nuanced features to light.” -the word nuanced feels strange here.

Reply: We have deleted the second part of this sentence.

Point 1.10 — “Simpler representations can be formed by removing “unknowable” nodes (Fig. 5B)” -unknowable is vague here, do you mean bubbles along a single lineage?

Reply: We've added a clarification that this refers to nodes such as those in singly-connected graph components.

Point 1.11 — “The gARG encoding leads to highly efficient storage and processing of ARG data, ”-As gARG has various levels of precision, perhaps this needs to state that the ”gARG encoding can lead to...” or be more precise that this is a reduced precision level.

Reply: Amended as suggested to add “can lead to”.

Point 1.12 — “The succinct tree sequence data structure (usually known as a “tree sequence” for brevity) is a practical gARG implementation focused on efficiency.” - If the tree sequence is focused at a particular level of gARG simplification be precise about this.

Reply: We have left this sentence as is, since the tree sequence structure can record gARGs at various levels of simplification.

Point 1.13 — “Methods targeting large-scale datasets tend to simplify the inference problem by making a single, deterministic best-guess ” –I think this is the best guess of the topology, and the uncertainty in times given the ARG is downstream of this. If so please clarify. Also I'd perhaps explicitly acknowledge Deng et al (SINGER), e.g. “deterministic best-guess of the topology (see Deng et al for parallel developments addressing uncertainty with somewhat small sample sizes)” or something like that. While these deterministic approaches are a strong way forward for human biobank scale data, it's good to be highlight parallel developments that might be key to other applications.

Reply: We have mentioned this as suggested.

June 4, 2024

RE: GENETICS-2024-307152

Dr. Jerome Kelleher
University of Oxford
Big Data Institute
Roosevelt Drive
Oxford, N/A
United Kingdom

Dear Dr. Kelleher:

Congratulations! We are delighted to inform you that your manuscript entitled "A general and efficient representation of ancestral recombination graphs" is acceptable for publication in GENETICS. Many thanks for submitting your research to the journal, and your work on the resubmissions.

To Proceed to Production:

1. Format your article according to GENETICS style, as discussed at <https://academic.oup.com/genetics/pages/general-instructions>, and upload your final files at <https://genetics.msubmit.net>.
2. Your manuscript will be published as-is (unedited-as submitted, reviewed, and accepted) at the GENETICS website as an Advanced Access article and deposited into PubMed shortly after receipt of source files and the completed license to publish. Please notify sourcefiles@thegsajournals.org if you do not wish to publish your article via Advanced Access.
3. We invite you to submit an original color figure related to your paper for consideration as cover art. Please email your submission to the editorial office or upload it with your final files. You can submit a small-sized image for evaluation, and if selected, the final image must be a TIFF file 2513px wide by 3263px high (8.375 by 10.875 inches; resolution of 600ppi). Please avoid graphs and small type.

If you have any questions or encounter any problems while uploading your accepted manuscript files, please email the editorial office at sourcefiles@thegsajournals.org.

Sincerely,

Graham Coop
Associate Editor
GENETICS

Approved by:
Hongyu Zhao
Senior Editor
GENETICS

note: Please add jnls.author.support@oup.com and genetics.oup@kwgglobal.com (or the domains @oup.com and @kwgglobal.com) to your email program's "safe senders" list. You will be contacted by both at various points during the production process.

Review comments (if applicable):